



# A new parameterization of ice heterogeneous nucleation coupled to aerosol chemistry in WRF-Chem model version 3.5.1: evaluation through the ISDAC measurements

Setigui Aboubacar KEITA[1], Eric GIRARD[1,†], Jean-Christophe RAUT[1,2], Maud LERICHE[1,3], Jean-Pierre BLANCHET[1], Jacques PELON[2], Tatsuo ONISHI[2], and Ana CIRISAN[1]

[1]ESCER Centre, Department of Earth and Atmospheric Sciences, Université du Québec à Montréal, H3C 3P8, Montréal, Québec, Canada
[2]LATMOS/IPSL, Sorbonne Université, UVSQ, CNRS, Paris, France
[3]Laboratoire d'Aérologie (LA), CNRS, Université Paul Sabatier, Toulouse, France
[†]deceased, 10 July 2017

**Correspondence:** Setigui Aboubacar KEITA (keita.setigui_aboubacar@courrier.uqam.ca)

**Abstract.** In the Arctic, during polar night and early spring, ice clouds are separated into two leading types: (1) TIC1 clouds characterized by large concentration of very small crystals, and TIC2 clouds characterized by low concentration of large ice crystals. Using suitable parameterization of heterogeneous ice nucleation is essential for properly representing ice cloud in meteorological and climate model and subsequently understanding their interactions with aerosols and radiation. Here, we describe a new parameterization for ice crystals formation by heterogeneous nucleation coupled to aerosols chemistry in WRF-Chem. The parameterization is implemented in the Milbrandt and Yau's two-moment cloud microphysics scheme and we assess how the WRF-Chem model responds to the real time interaction between chemistry and the new parameterization. Well-documented reference cases provided us in situ data from the spring 2008 Indirect and Semi-Direct Aerosol Campaign (ISDAC) campaign over Alaska. Our analysis reveals that the new parameterization clearly improves the representation of the IWC in polluted or unpolluted air masses and shows the poor performance of the reference parameterization in representing ice clouds with low IWC. The new parameterization is, thus able to represent TIC1 and TIC2 microphysical characteristics at the top of the clouds were heterogeneous ice nucleation is most likely occurring even knowing the bias of simulated aerosols by WRF-Chem over Arctic.

## 1 Introduction

The Arctic is warming faster than the global mean, and projections for the future suggest that this tendency will continue (Intergovernmental Panel on Climate Change) (IPCC, 2013). The contribution of aerosols to the changing climate of the Arctic is poorly known. Aerosols perturb the radiative balance directly by absorbing radiation and indirectly due to aerosol effects on clouds properties leading to increases in shortwave scattering efficiency and IR emissivity alterations of Arctic clouds (Zhao and Garrett, 2015; Shindell and Faluvegi, 2009). The radiative properties and lifetime of clouds are particularly sensitive to aerosol concentration, composition and size. While the uncertainties associated with the indirect effects of aerosol on liquid





clouds are still large, the effect of ice nucleation is even less well understood. Ice particle formation in tropospheric clouds significantly changes cloud microphysical properties, radiation balance and precipitation efficiency. At the core of the problem, ice nucleation causes multiple changes to clouds behavior, which at present are difficult to quantify. In its latest report, the IPCC, was unable to estimate the radiative forcing of aerosol on clouds through ice nucleation (Boucher et al., 2013).

Efforts are still needed to understand fundamentals processes of ice nucleation in clouds to improve their parameterizations, which are are particularly difficult, given the paucity of observations specifically in Artic (Curry et al., 1996; Kanji et al., 2017; McFarquhar et al., 2017). Instead of using assumptions, such as, for instance, that ice particles and cloud droplets are spatially homogeneously distributed; using parametrization based upon observations may be an alternative to reduce model uncertainties, (Kay et al., 2016). Central to the problem, the efficiency of ice nuclei (IN) to nucleate via freezing processes can

be described either through the stochastic approach or through the singular approach (Connolly et al., 2013; Niedermeier et al., 2011). In the singular (deterministic) approach, ice nucleation occurs at fixed temperature and humidity conditions assuming a characteristic number density of surface sites on aerosol particles. Ice nucleation in the stochastic approach is time dependent and is described by the classical nucleation theory (CNT) (Pruppacher and Klett, 1998) . In this approach, freezing occurs at any location on the micro-surface of a particle with equal probability. The best approach is still a matter of debates (Vali, 2014;

Wright and Petters, 2013).

Most of atmospheric models use simple time-dependent parameterization of ice nucleation predicting ice crystal number concentration, either as a function of temperature (Fletcher, 1962; Cooper, 1986) or ice supersaturation (e.g., (Meyers et al., 1991)). These parameterizations do not include a limitation of ice crystal number concentration by the number of available ice nuclei particles and can lead to very poor estimation of ice crystal number concentration, in particular, if they are applied

outside of the range of measurements used to constrain them (Prenni et al., 2007). This is particularly true for ice clouds in Arctic conditions (Keita and Girard, 2016). In the CNT model case using a fitting parameter, the contact angle ($\theta$), which can be described as a single-contact-angle for an entire population does not work well for predicting the fractions of ice nuclei (IN) on dust aerosol or on particles that have heterogeneous surfaces (Hoose and Möhler, 2012).

In recent years, with increasing data on ice nucleation from field and laboratory studies, new time-independent parameteri-

zations have been developed, often based on empirical fits to atmospheric IN measurements as a function of temperature and aerosol particle size distributions (e.g., (Connolly et al., 2013; Welti et al., 2012; Phillips et al., 2013; DeMott et al., 2010, 2015; Cirisan et al., 2019)). Despite significant advances, they are of limited use in large-scale models operating over a wide range of temperatures. More complex CNT parameterizations than those using contact angle ($\theta$-PDF) come at high computational costs (Welti et al., 2012; Murray et al., 2012; Niedermeier et al., 2014). In the particular context of climate simulations under

Arctic atmospheric and chemical conditions, there is a need for efficient parameterizations of heterogeneous ice nucleation using simplified approaches to limit computational time.

In (Keita et al., 2019) the parameterization of (Girard et al., 2013) based upon CNT approach was implemented in the online WRF-Chem chemistry-transport model (Grell et al., 2005). This parameterization assumed that IN are mainly mineral dust particles, which is consistent with recent results from the NETCARE project (Abbatt et al., 2019). This parameterization con-

sidered physico-chemical properties of IN, important in Arctic conditions especially during winter and early spring (Eastwood





et al., 2009; Keita and Girard, 2016) when sulfuric acid is often a dominant component of the aerosol, known as arctic haze. Two Types of Ice Clouds (TICs) had been characterized (Grenier et al., 2009). A TIC1 cloud is composed by a relatively large number of non-precipitating small ice crystals, set to less than 30 $\mu$m in diameter. The second type, TIC2 cloud, is characterized by a low concentration of larger precipitating ice crystals (diameter larger than 30 $\mu$m). After spatial and temporal evaluation

of the model, (Keita et al., 2019) showed the ability of the parameterization to discriminate TIC1 and TIC2 clouds observed during Indirect and Semi-Direct Aerosol Campaign (ISDAC), (McFarquhar et al., 2011). However, the study of (Keita et al., 2019) was constrained by a prescribed concentration of aerosols with fixed acid concentration.

In this paper, we investigate for the first time ice nucleation in a fully coupled aerosol and chemistry parameterization. We evaluate the response of the WRF-Chem model to the realistic time dependent interaction between aerosols, predicted by

the chemistry module, and the contact angle approach proposed by (Girard et al., 2013). The new parameterization improves significantly the treatment of ice nucleation by discriminating TIC1 and TIC2 clouds formation as a function of the aerosol chemical composition. Each cloud is closely analyzed against observations from three detailed flights data taken during ISDAC (2008). This study is a part of the NETCARE project addressing key uncertainties in Remote Canadian Environments with the objectives of assessing the impact of aerosols on Arctic ice clouds.

The paper is organized as follows. Section 2 briefly describes the (Milbrandt and Yau, 2005a, b) scheme for cloud microphysics and the full presentation of ice heterogeneous nucleation parameterization coupled with aerosol chemistry. Section 3 presents the test cases from the ISDAC campaign and section 4 the evaluation of the new parameterization against the ISDAC campaign. Section 5 is dedicated to conclusion.

## 2 Description of the new scheme for ice heterogeneous nucleation in WRF-Chem

The new scheme for ice crystals formation by heterogeneous nucleation is implemented in WRF-Chem Version 3.5.1. WRF-Chem is a regional, fully-coupled "online" model (Grell et al., 2005), where all prognostic meteorological, chemical and aerosol variables are fully integrated within WRF-ARW, a mesoscale meteorological model, and uses the same grid, time step, advection scheme and physics schemes as WRF-ARW. Several schemes are available in WRF-Chem for cloud microphysics. We choose the (Milbrandt and Yau, 2005b) – MY05, for its ability to simulate Arctic clouds in previous works (Keita et al.,

2019; Keita and Girard, 2016).

### 2.1 Overview of the two-moment version of the cloud microphysical scheme MY05

MY05 (Milbrandt and Yau, 2005a, b) is a bulk cloud microphysics parameterization with one, two and three-moment versions. We use the two-moment version available in WRF-Chem. It includes the following prognostic variables: the mass mixing ratio ($q_x$) and the number concentration ($N_x$) with x∈ (c, r, i, s, h, g) representing respectively cloud liquid water (c), cloud ice water

(i), rain (r), snow (s), hail (h) and graupel (g). All symbols for variables and parameters used are listed in Table 1. The time





evolutions of hydrometeor mass mixing ratio and number concentration are respectively, governed by the following prognostic equations:

$$\frac{\partial q_x}{\partial t} = -\frac{1}{\rho}\nabla(\rho q_x) + TURB(q_x) + \frac{1}{\rho}\frac{\partial}{\partial t}(\rho q_x V_{Qx}) + \frac{dq_x}{dt}|_s \tag{1}$$

and

$$\frac{\partial N_{Tx}}{\partial t} = -\nabla.(N_{Tx}U) + TURB(N_{Tx}) + \frac{\partial}{\partial z}(N_{Tx}V_{Nx}) + \frac{dN_{Tx}}{dt}|_s \tag{2}$$

where $\rho$ is the density of air, $\mathbf{U}$ is the 3D velocity vector, $(V_{Qx})$ is the mass weighted fall speed, $(N_{Tx})$ is the total number concentration per unit volume and $(V_{Nx})$ is the number weighted fall speed. The terms on the right of both equations represent, respectively, advection/divergence, turbulent mixing, sedimentation, and microphysical tendencies (marked by s subscript). The mass of a single hydrometeor for the x category is parameterized as a power law of the form:

$$m_x(D) = c_x D^{d_x} \tag{3}$$

where $d_x = 3$ for all hydrometeors and $c_x = \rho_x \frac{\pi}{6}$ , with $\rho_x$ being the bulk density (Table 2) for spherical particles x (cloud liquid water, rain, snow, graupel, and hail). Cloud ice crystals are assumed to be bullet rosettes (Ferrier, 1994) with $c_i = 440$ kg m$^{-3}$. The size spectrum of each category is described by a common generalized gamma distribution function (Cohard and Pinty, 2000; Ferrier, 1994) of the form:

$$dN_x(D) = N_{Tx}(D)\frac{\nu_x}{\Gamma(1+\alpha_x)}\lambda_x{}^{\nu_x(1+\alpha_x)}D^{\nu_x(1+\alpha_x)-1}\exp[(-\lambda_x D^{\nu_x})] \tag{4}$$

where $dN_x(D)$ is the number concentration of hydrometeor x per unit volume per unit diameter D, $(\alpha_x)$ is the shape parameter controlling the size dispersion, $(\lambda_x)$ is the slope and $(\nu_x)$ is a second size dispersion parameter. The size distribution of cloud droplets is represented in MY05 by $(\alpha_x)= 1$ and $(\lambda_x) = 3$. For all other hydrometeors $(\nu_x)=1$ leading to the form:

$$dN_x(D) = N_{0x}D^{\alpha_x}\exp[(-\lambda_x D)] \tag{5}$$

where $N_{0x}$ is the intercept parameter given by:

$$N_{0x} = N_{Tx}(D)\frac{1}{\Gamma(1+\alpha_x)}\lambda_x{}^{(1+\alpha_x)} \tag{6}$$

The four ice-phase hydrometeors follow the above size distribution. The cloud ice water category represents pristine ice crystals. The snow category includes crystals with radii greater than 100 μm and aggregates. The graupel category includes





moderate-density graupels, formed from heavily rimed ice or snow. The hail category corresponds to high-density hail and

frozen raindrops. For each ice-phase hydrometeor x, the total number concentration $N_{T,x}$ (kg$^{-1}$) and the mass mixing ratio $q_{T,x}$ (kg kg$^{-1}$) is given respectively by:

$$N_{T,x} = \int_0^\infty N_{0x}D^{\alpha_x}\exp(-\lambda_x D)dD \tag{7}$$

and

$$q_{T,x} = \frac{1}{\rho}\int_0^\infty m_x(D)N_{0x}D^{\alpha_x}\exp(-\lambda_x D)dD \tag{8}$$

where $m_x(D)$ is obtained from Eq. (3).

Microphysical processes represented in MY05 are summarized in Table 3, where processes are listed according to the hydrometeor category. The source and sink terms for the two-moment (mass content) are from previous studies (Cohard and Pinty, 2000; Kong and Yau, 1997; Ferrier, 1994) and depend on the size distribution function. The primary sources of ice crystals in the atmosphere are the heterogeneous and homogeneous ice nucleation. Homogeneous freezing is the spontaneous

freezing of a water (or haze) droplet. According to (Pruppacher and Klett, 1998), the homogeneous freezing rate of cloud droplets is dominant at temperatures below $\sim -32°C$. In the range $-30°C$ to $-50°C$, MY05 follows (DeMott et al., 1994) with:

$$\Delta N_{freeze} = \int_0^\infty [1-\exp(-JV\Delta t)]N_{Tc}(D)dD \tag{9}$$

In a given time step ($\Delta t$), ($\Delta N_{freeze}$) is the number of droplets that freezes by homogeneous freezing and ($J$) is the

nucleation rate for pure water. For homogeneous nucleation:

$$\log_{10}(J) = -606.3952 - 52.6611T_c - 1.7439T_c^2 - 2.65\times10^{-2}T_c^3 - 1.536\times10^{-4}T_c^4 \tag{10}$$

with the volume $V$ approximated by the mean-droplet diameter in units of cm. Therefore, the fraction of cloud droplets freezing in one time step may be written as:

$$F_{freeze} = \frac{\Delta N_{freeze}}{N_{Tc}}[1-\exp(-J\frac{\pi}{6}D_{mc}^3\Delta t)] \tag{11}$$

where $D_{mc}$ is the mean volume diameter of cloud droplets. Heterogeneous ice nucleation needs ice nuclei (IN), a minor fraction of the tropospheric aerosol, which exhibits micro surface structures to facilitate the formation of ice crystals. In





presence of IN, if thermodynamic conditions are favourable, ice crystals can form by heterogeneous nucleation through four different modes. Deposition nucleation and condensation freezing can occur without the presence of supercooled droplets. For clouds below $0°C$, primarily composed of supercooled liquid droplets, ice crystal can form by immersion and contact freezing. This conceptual definition of heterogeneous ice nucleation (Pruppacher and Klett, 1998) is used in MY05. Contact freezing follows (Young, 1974) where the number concentration of contact IN is a function of temperature according to (Meyers et al., 1991). In the contact freezing formation mode, ice nucleation occurs on a solid particle colliding with a supercooled liquid droplet. Immersion freezing of raindrops and cloud water droplets follows the parameterization of (Bigg, 1953). The deposition mode involves the growth of ice directly from the vapour phase, whereas condensation freezing occurs if the ice phase is formed immediately after condensation of water vapor on a solid particle as liquid intermediate. In the original version of MY05, deposition and condensation-freezing are functions of water vapour supersaturation with respect to ice, $S_i$, following (Meyers et al., 1991):

$$N_{m,i}(S_i) = 1000 \exp[12.96(S_i - 1) - 0.639] \tag{12}$$

where $N_{m,i}$ is the number of ice crystals predicted per unit volume due to deposition and condensation-freezing. The (Meyers et al., 1991) parameterization for deposition and condensation freezing depends only on supersaturation. It was derived from ground-based measurements. These approximations may lead to an overestimation of $N_i$ when the number concentration of particles acting as IN is low, such as in Arctic conditions (Eidhammer et al., 2009) . Moreover, the immersion freezing mode from (Pruppacher and Klett, 1998) has been extended to include freezing of immerged IN inside an aqueous solution or wet aerosol (Vali et al., 2015), which is a significant process of Arctic ice clouds formation (Eastwood et al., 2008).

### 2.1.1 A new parameterization of ice heterogeneous nucleation coupled with chemistry for MY05 in WRF-Chem

The new parameterization focuses on deposition ice nucleation for uncoated IN and to immersion freezing of sulphuric acid coated IN, i.e. IN immerged in an acid aqueous solution. In this approach, IN are assumed to be mineral dust particles following (Girard et al., 2013). For contact freezing and immersion freezing from supercooled cloud droplets, the parameterizations remain unchanged. For condensation-freezing, it can be included in the immersion freezing of coated IN when air is supersaturated with respect to liquid water. However, as discussed in (Vali et al., 2015) , this process is uncertain. The modified version of MY05 including our new parameterization described below is referred hereafter to MYKE. The parameterization is based on the CNT, a stochastic approach in which the nucleation rate $J_d$ depends on the contact angle between an ice embryo and its IN. Following CNT, in each time step ($\Delta t$) the number concentration of nucleated ice crystals ($N_f$) is given by:

$$N_f(\Delta t) = N_0 \exp[1 - J_d A_d \Delta t] \tag{13}$$

where $A_d$ is the surface area of dust particles and $N_0$ is the total number concentration of available IN. In previous studies, using this approach (Keita and Girard, 2016; Keita et al., 2019; Girard et al., 2013; Khvorostyanov and Curry, 2009; Morisson





et al., 2005; Liu et al., 2007; Hoose et al., 2010; Chen et al., 2008), $A_d$ and $N_0$ were prescribed and constant over time although the concentration of atmospheric IN varies tremendously in time and space, as well as in their composition and origins. The new MYKE parameterization within WRF-chem, now considers the temporal and spatial variation of $A_d$ and $N_0$. $J_d$, the
nucleation rate of embryos per unit surface of particles (Pruppacher and Klett, 1998; Martin, 2000; Hung et al., 2003; Parsons et al., 2004a, b; Archuleta et al., 2005; Pant et al., 2006), is defined as:

$$J_d(cm^{-2}s^{-1}) = B\exp(\frac{\Delta G^*}{kT}) \tag{14}$$

where $B$ is the pre-exponential factor (Pruppacher and Klett, 1998) function of the aerosol particle (nucleus) mean radius $r_n$ defined as:

$$B(cm^{-2}s^{-1}) = 10^{-26}r_n^2 \tag{15}$$

where $k$ is the Boltzmann constant in $JK^{-1}$, $T$ is the temperature in K, $\Delta G^*$ is the critical Gibbs free energy for the formation of an ice embryo in J and is defined as:

$$\Delta G^* = \frac{16\pi\sigma_{iv}^3 f(\cos\theta)}{3\rho_i^2 R_v^2 T^2 \ln^2 S_i} \tag{16}$$

where $\sigma_{iv} = 06.510^3 J.m^{-2}$ is the surface tension between ice and water vapour, $\rho_i = 0.5g.cm^{-3}$ is the bulk ice density,
$R_v = 461.5 J.kg^{-1}K^{-1}$ is the gas constant for water vapor. The function $f(\cos\theta)$ is a monotonic decreasing function of the cosine of the contact angle $\theta$ as defined by (Pruppacher and Klett, 1998) for an infinite plane surface:

$$f(\cos\theta) = \frac{1}{2}\left\{1 + \left(\frac{1-q\cos\theta}{\phi}\right)^3 + q^3\left[2 - 3(\frac{q-\cos\theta}{\phi}) + \left(\frac{q-q\cos\theta}{\phi}\right) + 3q^2\cos\theta\left(\frac{q-q\cos\theta}{\phi} - 1\right)\right]\right\} \tag{17}$$

where $\phi = \sqrt{1 - 2q\cos\theta + q^2}$ and $q = \frac{r_n}{r_g}$ with $r_g$ being the critical germ size expressed as:

$$r_g = \frac{2\nu_w \sigma_{iv}}{kT\ln(S_i)} \tag{18}$$

where $\nu_w$ are the volume of a water molecule. In the CNT, the contact angle $\theta$ is a very important variable because it represents the ability of an IN to form ice. The lower the contact angle, the better IN the aerosol is. Numerous laboratory studies have found realistic values of $\theta$ based on the physicochemical composition of aerosols (e.g., (Marcolli et al., 2007; Eastwood et al., 2008; Fornea et al., 2009; Welti et al., 2009; Kanji and Abbatt, 2010; Kulkarni and Dobbie, 2010; Welti et al., 2009)). The CNT approach using these values was subsequently applied successfully in climate and forecast models at different
scales (Khvorostyanov and Curry, 2009; Morrison and Curry, 2005; Liu et al., 2007; Chen et al., 2008). For example, (Keita





et al., 2019) using the parameterization of (Girard et al., 2013) based on laboratory studies from (Eastwood et al., 2008, 2009) were able to simulate Arctic clouds forming in polluted and clean air masses with a prescribed contact angle of 26° and 12° respectively. These studies were, however, limited on the one hand because the contact angles represent extreme cases that must be prescribed arbitrarily before the simulation and, on the other hand, they assumed homogeneity of the degree of acidity

of clouds in space and in time throughout the whole domain. For the first time, here a real-time variable contact angle is used in the CNT approach by coupling MY05 with the chemical module in WRF-Chem. This coupling is between MY05 and the MOSAIC (Model for Simulating Aerosol Interactions and Chemistry) aerosol module (Zaveri et al., 2008). MOSAIC simulates a wide variety of aerosol species: sulphates, methanesulfonate, nitrate, chloride, carbonate, ammonium, sodium, calcium, black carbon (BC), primary organic mass (OC), liquid water, and other inorganic mass (OIN). OIN represents unspecified inorganic

species such as silica ($SiO2$), other inert minerals, and trace metals, lumped together assimilated to mineral dusts. MOSAIC uses a sectional approach to represent aerosol size distributions by dividing up the size distribution for each species into several size bins (4 or 8 available in WRF-Chem) and assumes that the aerosols are internally mixed in each bin. MOSAIC considers major aerosol processes: inorganic aerosol thermodynamic equilibrium, binary aerosol nucleation, coagulation and condensation, but does not include the secondary organic aerosol (SOA) formation in the version used in this study. MOSAIC is

a good compromise between accuracy and computing performance. It is used in WRF-Chem with four chemical mechanisms. The coupling is done by expressing $\theta$ as function of the neutralized fraction ($f$) in aerosol particles (Zhang et al., 2007; Fisher et al., 2011), which is between 0 and 1 and is defined as:

$$f = \frac{\left[NH_4^+\right]}{2\left[SO_4^{2-}\right] + \left[NO_3^-\right]} \tag{19}$$

This was motivated by several previous studies (Jouan et al., 2012; Grenier and Blanchet, 2010; DeMott et al., 2010; Girard

and Blanchet, 1994; Keita et al., 2019; Keita and Girard, 2016) suggesting that the acidification of ice nuclei by the oxidation of sulphur dioxide forming sulphuric acid in Arctic greatly alters the microphysical response of ice clouds. Such ice clouds tend to have bigger and fewer ice crystals than ice clouds formed in pristine environments. Moreover, $\theta$ has been derived by (Eastwood et al., 2008, 2009) from heterogeneous nucleation rates obtained in laboratory measurements. As best fit, they found limiting values of $\theta = 26°$ in polluted air and $\theta = 12°$ in clean air. (Keita and Girard, 2016) , after analysed the slope between the

nucleation rate and the saturation over ice for TIC1 and TIC2 clouds (cf. Fig. 16 in (Keita and Girard, 2016)) observed for a given Si that: (1) the slope is the largest for the smallest accessible contact angle; (2) the decrease of the slope with the increase of contact angle is very non-linear. These results are consistent with laboratory experiments (Sullivan et al., 2010) showing a rapid increase of contact angle with acidity on coated IN. These results motivate us to parameterize the contact angle $\theta$ (in degrees) as function of the neutralized factor either in a quadratic or a biquadratic form:

$$\theta = 26 - 14f^2 \tag{20}$$





$$\theta = 26 - 14f^4 \tag{21}$$

Both formulations are implemented in MY05 and tested hereafter. They imply that $\theta$ is close to 26° for 0 < f < 0.5 with a more (21) or less (20) rapid decrease between 0.5 and 1 as shown in Fig. 1. The coupling between MY05 and MOSAIC is done by taking information from MOSAIC for $A_d$ and $N_0$ as needed to compute Eq. (13); for $Ar_n$ to compute Eq. (15; 17) and for

$f$ to compute Eq. (20; 201). These parameters are computed assuming the same aerosol size bin definition as in MOSAIC.

## 3   Configuration of the model for typical TIC1 and TIC2 clouds observed during the ISDAC campaign

The ISDAC campaign took place during April 2008 at the North Slope of Alaska. The objective was to study the role of Arctic aerosols on cloud microphysical properties and on the surface energy budget. Numerous studies have been based upon data from the ISDAC campaign (McFarquhar et al., 2011; Lawson et al., 2019). Among them, several studies investigated detailed

parameters of ice clouds by analysing the ISDAC database (Jouan et al., 2012; Grenier and Blanchet, 2010; DeMott et al., 2010) or by running atmospheric models on case studies highlighted during the campaign (Keita and Girard, 2016; Matrosov et al., 2019; Keita et al., 2019). For instance, (Keita et al., 2019) analysed microphysical properties of TICs for ISDAC flights in non-polluted and polluted environment using WRF simulations. Flights F13 on the one side and F21 and F29 on the other side studied by (Keita et al., 2019) were typical of a TIC1 cloud formed in a pristine air mass and of two TIC2 representative

cloud cases formed in a polluted air mass, respectively. Here, our goal is to show the potential of the new ice nucleation parameterization to discriminate TIC1 and TIC2 clouds formation as a function of the aerosol chemical composition. Each cloud types are closely investigated using detailed observations from three flights taken during ISDAC. The simulations with WRF-Chem including MYKE are done over the whole period of the ISDAC campaign (McFarquhar et al., 2011), from April 1 to 30, 2008, on the domain shown in Fig. 2, and identical to that described by (Keita et al., 2019). The three test cases (F13, F21

and F29) are included in this period. The domain is based on a Lambert projection centred on Barrow, Alaska over 160 × 100 grid cells with a horizontal resolution of 10 km and 55 vertical levels between the surface and 50 hPa. The first 4 days of the simulation (1 to 4 April included) are used for model spin-up. Three simulations are performed: the first one uses the original MY05 scheme (the REF simulation), the second one uses the new parameterization given in Eq. (20a) (the MYKE2 simulation) and the third one uses the new parameterization described by Eq. (20b) (the MYKE4 simulation). WRF-Chem options and

parameterizations used in these simulations are summarized in Table 4. As in (Keita et al., 2019), meteorological initial and boundary conditions use NCEP (National Centers for Environmental Prediction) Global Forecast System (GFS) Final Analysis (FNL) data (1° x 1°) and the simulations are nudged to GFS-FNL updated every 6 hours above the planetary boundary layer (PBL). For the chemical module, the CBM-Z (Carbon Bond Mechanism) photochemical mechanism (Zaveri and Peters, 1999) coupled with MOSAIC is used. CBMZ has 67 species and 164 reactions in a lumped structure approach that classifies organic

compounds according to their internal bond types. Rates for photolytic reactions are derived using the Fast-J photolysis rate scheme (Wild et al., 2000). Eight size bins are used in MOSAIC. Chemical initial and boundary conditions are taken from the





global chemical-transport model MOZART-4 (Model for OZone And Related chemical Tracers, version 4) (Emmons et al., 2010). The fire emissions inventory used is the Fire INventory from NCAR (FINN-v1) (Wiedinmyer et al., 2011). FINN-v1 provides emissions on a per fire basis based on event count information from the MODIS (Moderate Resolution Imaging

Spectrometer) instrument. The anthropogenic emissions come from the inventory developed within the POLARCAT Model Intercomparison Model Project (POLMIP), which includes $SO_2$ from both eruptive and non-eruptive continuous degassing volcanism (Fisher et al., 2011; Jouan et al., 2014). During winter and spring 2008, sustained eruptive activity was recorded at the Kamchatka and the Aleutian Islands (Fisher et al., 2011; Jouan et al., 2014; Atkinson et al., 2013; Burton et al., 2012). Non-eruptive activity was common throughout our simulation period (Fisher et al., 2011; Jouan et al., 2014; Atkinson et al.,

2013). Soil-derived (dust) and sea salt aerosol emissions are computed online into WRF-Chem based upon, respectively, the wind erosion formulation of Shaw et al. (2008) and the GOCART (Global Ozone Chemistry Aerosol Radiation and Transport model) sea salt emission module (Chin et al., 2000). For biogenic emissions, the Model of Emissions of Gases and Aerosols from Nature (MEGAN) (Guenther, 2007) compute them online using characteristics of the surface (class of vegetation, soil humidity and temperature for instance).

## 260 4  Results and discussion

This section is dedicated to present comparisons of WRF-Chem simulations (REF, MYKE2 and MYKE4) against observations, followed by a discussion of the results. Although the comparison between simulated results and observations are presented in the following along the entire vertical profile inside the clouds, the discussion focuses on the altitudes above the 500 hPa level, where heterogeneous nucleation is the most important process. According to (Jouan et al., 2012), most of the differences

between TIC1 and TIC2 events were confined at cloud top where ice nucleation mostly occurs, and air is supersaturated with respect to ice. To compare simulations with observations along the ISDAC flight tracks, simulated results are averaged in a grid box of 10 by 10 km centred on the location of the flight. ISDAC in situ measurements have been averaged every 20 seconds, corresponding to a vertical resolution of $\sim$45 hPa ($\sim$450 m), during ascents and descents of the flight through clouds. Simulated WRF outputs are linearly interpolated to the pressure levels of these observations and temporally averaged over a

three hours period encompassing the area of ISDAC flights. Some statistics are computed using the same method. First, we present some meteorological and chemical properties followed by analyse of cloud microphysical properties.

### 4.1  Temperature and relative humidity over ice

Table 5 presents biases (Bias), correlation coefficients (Cor) and root mean square errors (RMSE), for the temperature (T) and relative humidity over ice ($RH_i$) for the three simulations (REF, MYK2 and MYKE4) and above the 500 hPa level. According

to (Jouan et al., 2012), the uncertainties on the measurements are estimated at $\pm$0.5°C for T and $\pm$11% for ($RH_i$). Note that, vertical profiles of T and ($RH_i$) for F13, F21 and F29 flights are very close to results obtained by (Keita et al., 2019). As expected, due to the nudging, the new heterogeneous ice nucleation parametrisation does not significantly impact T and RHi. The lowest temperatures at the top of the clouds, where the process of heterogeneous ice nucleation is important, are relatively





well reproduced by MYKE2 and MYKE4 simulations with similar statistics (Cor≃0.99, RMSE≃2, Bias≃-2°C), except along
F21 flight (Cor≃0.82, RMSE≃3.3, Bias≃-3°C), where the observed increasing of temperature caused by the heat exchanged
at cold temperatures is not adequately represented by the model. For that flight, the three simulations underestimate RHi by
±50% at the top of the cloud. These biases are consistent with the large-scale GFS-FLN fields and results in an underestimation
of the altitude of the top of the cloud by the model for F21.

## 4.2    Aerosol properties

Figure 3 shows the comparison between observed and simulated (REF, MYKE2 and MYK4) vertical profiles of aerosol num-
ber concentrations (Na). The Passive Cavity Aerosol Spectrometer Probe (PCASP) externally mounted under a wing of the
Convair-580 aircraft sampled ambient clear air just before entering the cloud regions for all flights except F21. The optical
particle counter (PCASP) provided particle size distributions and number concentrations in the geometric diameters size range
0.12 – 3 μm. To allow a fair comparison between WRF-Chem simulated and PCASP-measured Na, the model concentrations
are summed over bins 3 to 6, corresponding to sizes between 0.156 and 2.5 $\mu m$. According to (Shantz et al., 2014), the un-
certainty in number concentration measured by the PCASP is approximately 10%. First, the model does not reproduce the
observed vertical variability. It maybe due to the small sampling domain and time taken during ISDAC, which make compar-
isons between model simulations and the observed variability difficult, especially at the low horizontal resolution of 10 km
used here. For F13, the air mass is relatively clean with a weak vertical variability of aerosol number concentrations, remaining
mostly below 210 $cm^3$ on the whole column with mean concentrations around 73 $cm^3$, very close to simulation mean 86 $cm^3$.
For F29, the PCASP instrument show that there is a much higher concentration of aerosol particles in the lower troposphere
(more than twice that observed during F13, e.g., larger than 400 $cm^3$ and particularly at altitudes above 550 hPa, near cloud
top where peak concentrations exceeding 1000 $cm^3$ have been measured. Comparing the two flights, between 550 hPa and
400 hPa, the simulated aerosol number concentration is overestimated by 3 against observations for F13 flight and is underes-
timated by one order magnitude for F29 flight (Fig. 3). These discrepancies are consistent with (Mölders et al., 2011), which
analysed aerosols concentration during polar night around Fairbanks, and showed an overestimation of aerosol concentration
over the non-polluted site and an underestimation on polluted site by using WRF-Chem. They concluded that discrepancies
result from uncertainty in emissions especially at Fairbanks. While most models agree that Arctic aerosols can be attributed to
a mixture of anthropogenic sources, meso-scale models have difficulty to simulate properly aerosol concentrations over Arctic
(Shindell and Chin, 2008; Eckhardt et al., 2015; Schwarz et al., 2013; Raut et al., 2017). Moreover, even if the simulated results
show the same order of magnitude for Na above 550 hPa (Fig. 3) whereas observations show a large difference between the
two flights, we expect that the differences between simulated results for cloud microphysical properties for these two flights
could be mainly explained by a combination of differences of the physico-chemical properties of aerosols and of the altitude
of the simulated cloud top. Figure 4 presents simulated (REF, MYKE2 and MYKE4) vertical profiles of respectively sulphate
($SO_2$), ammonium ($NH_4$) and nitrate ($NO_3$) molar aerosol concentrations along the flights F13, F21 and F29. Unfortunately,
no observation of the aerosol chemical composition was available during the campaign to evaluate those results. Vertical distri-
butions indicate a rather constant structure of aerosol molar concentrations for F13 with mean value around 6.2 $nmol/cm^3$ for





both SO4 and NH4, and 0.5 $\mathrm{nmol/cm}^3$ for ($NO_3$ ) (Fig. 4). For F21 and F29 simulated results show peak aerosols concentrations in the mid-troposphere up to a factor 2 compared to F13, and a larger vertical gradient, with large and moderate depletion

in the boundary layer respectively for F21 and F29 (Fig. 4B and Fig. 4C). F21 and F29 have NH4 mean value respectively 8 and 10.2 $\mathrm{nmol/cm}^3$ and SO4 mean value both around 7 $\mathrm{nmol/cm}^3$. These values and the vertical structures correspond relatively well to mean observed concentrations for $NH_4$ and $SO_2$ respectively 7 $\mathrm{nmol/cm}^3$ seen during ARCTAS (Arctic Research of the Composition of the Troposphere from Aircraft and Satellites) and ARCPAC (Aerosol, Radiation, and Cloud Processes affecting Arctic Climate) campaigns of April 2008 (Fisher et al., 2011). (Fisher et al., 2011) showed that volcanic

sources (Aleutian Islands and Kamchatka) accounted for $12 - 24\%$ of the sulphate at all altitudes, with peak contribution in the mid-troposphere. The volcanic source is discharged directly in the free troposphere and is thus less affected by deposition than surface sources. This is also supported by satellite observations from the Ozone Monitoring Instrument (OMI) over the North Slope of Alaska, which shows much larger $SO_2$ concentrations at the end of the ISDAC campaign. Clouds sampled during both F21 and F29 appear to form mostly in air masses containing dust and smoke, possibly with a highly acidic coating.

Figure 5 presents the vertical profile of the factor $f$ (full line, see Eq. 19) and the contact angle $\theta$ (dashed, see Eq. 20A, 20B) for MYKE2 (Figure 6A) and for MYKE4 (Figure 6B) along the top of the three flights F13, F21 and F29. Results obtained with the MYKE2 and MYKE4 are very similar with both using the same $f$. Results from the two simulations are therefore discussed together. The difference lies on the curve shape of the contact angle $\theta$, MYKE4 simulates a more rapid decrease between $0 < f < 0.5$ than MYKE2 (see Figure 1). This prescription substantially increases $\theta$ values in MYKE4 more than in

MYKE2 along the vertical profile by up to 3° especially at the cloud top where nucleation is the dominant process. This change has a positive impact on the nucleation rate: a smaller contact angle in the MYKE2 simulation indeed tends to decrease the critical Gibbs free energy to form ice embryos (Eq. 16), hence leads to a higher nucleation rate of ice crystals. The $\theta$ profile in F13 presents a constant shape with values around 17.5° and 20.5° respectively for MYKE2 and MYKE4. Focus on MYKE4 for F21, the large contact angle around 21° corresponds to acid IN, i.e. a smaller f than F13, and a decrease in the nucleation

rate. Although F29 also shows a significant acidity around 400 $\mathrm{hPa}$, (Fig. 4B) with higher concentrations of SO4 than F13, it tends to neutrality around 500 $\mathrm{hPa}$ in relation to the increase of ammonium at this altitude in comparison to higher altitudes and the negligible amount of nitrate in the upper part of the cloud (Fig. 4B and 4C). Our results reveal that the model broadly reproduces Na from the ground to 500 hPa level, but it has difficulty to represent Na in the upper part, even if observations and model results remain of the same order of magnitude. MYKE2 and MYKE4 simulations show higher $\theta$ values at clouds top

for F21 and F29 in comparison to F13, thus differencing the acidic to the nonacidic cases as expected. In the following section, we will examine the effect of interactive chemistry on the cloud microphysical variables.

### 4.3 Cloud microphysical structure

Details of the retrieval of cloud microphysical properties and associated uncertainties from the several cloud probes on board the Convair-580 aircraft are given in (Jouan et al., 2012). Figure 6 presents the comparison of the observed and simulated

(REF, MYKE2 and MYKE4) vertical profiles of IWC (uncertainties: $\pm75\%$) along the three flights. Observed IWC vertical profiles for F13 and F29 continuously decreasing between 800 $\mathrm{hPa}$ and 400 $\mathrm{hPa}$ with values in the range of $10^{-1}$ $\mathrm{kg/kg}$ to





$10^{-2}$ kg/kg. For flight F21, observed IWC shows a large variability in its vertical structure. IWC values simulated by both MYKE2 and MYKE4 are very similar, with a slight improvement for MYKE2 simulating more IWC. This agrees with the $\theta$ difference between MYKE2 and MYKE4 (Fig. 5). A smaller contact angle in the MYKE2 simulation tends to decrease the

critical Gibbs free energy to form ice embryos (Eq.16), hence leads to a higher nucleation rate of ice crystals and higher IWC. Both MYKE2 and MYKE4 broadly capture observed values with a low bias: +1.2 $10^{-2}$ g/kgand +8.1 $10^{-3}$ g/kg for F13; -3.2 $10^{-3}$ g/kg and -3.5 $10^{-3}$ g/kg for F21; -2.1 $10^{-3}$ g/kg and -8.1 10-3 $10^{-3}$ g/kg for F29 respectively. On the contrary, REF strongly underestimates IWC values with a negative bias of 0.01 g/kg for F13 and 0.03 g/kg for F29. Note that REF does not have any noticeable IWC cloud at these levels in flight F21. Figure 7 presents a comparison between observed and

simulated (REF, MYKE2 and MYKE4) vertical profiles of Ni (uncertainties: $\pm 50\%$ in the upper part of the cloud where the heterogeneous ice nucleation processes are dominant, above 500 hPa, during F13, F21 and F29 flights. The airborne ISDAC vertical profile for the TIC1 observed during F13 varies between 70 and 200 $L^{-1}$ and is rather constant with altitude. The REF simulation strongly underestimates Ni by two orders of magnitude corresponding rather to a TIC2. MYKE2 and MYKE4 reproduce well the observed Ni within the ranges of uncertainties while MYKE4 is slightly closer to observations with a bias of

25 $L^{-1}$. The TIC2 cloud type observed along F21 and F29 flight tracks is characterized by a small concentration of ice crystals ranging between 1 and 30 $L^{-1}$. For F21, while REF is not able to simulate a persistent cloud, both MYKE2 and MYKE4 show a cloud with Ni close to observations typical of TIC2 under 450 hPa in the range of incertitude $\pm 50\%$. As expected, due to the biases of temperature and relative humidity over ice, the model underestimates the cloud top altitude for F21. For F29, both MYKE2 and MYKE4 show an increase in Ni comparing to REF, which has the best statistics, while MYKE2 and MYKE4

simulations are overestimated by one order of magnitude. However, it is reasonably close to satellite observations as analysed by (Keita et al., 2019). Their analysis reveals a large discrepancy of Ni between ISDAC flights and satellite estimations. It is likely due to the small sampling domain taken during ISDAC versus the low resolution of satellite products and of the model grid. Figure 8 presents the comparison of the observed and simulated (REF, MYKE2 and MYKE4) vertical profiles of Ri with uncertainties of $\pm 97\%$) along the F13, F21 and F29 flights. Observations show that, although having the same IWC magnitude

(Figure 6), the TIC1 and TIC2 differ by their Ni (Figure 7) and the Ri values. F13 flight (TIC1) with large Ni concentration has Ri values around 25 $\mu$m while both F21 and F29 flights refer to TIC2 with low Ni and Ri at least a factor two larger. The IN acid coating in TIC2 inhibits the ice nuclei properties of the IN, slowing the rate of ice nucleation in comparison to uncoated Ni. Subsequently, this decrease of the nucleation rate increases the amount of available supersaturated water vapour and allows the rapid growth of activated ice crystals. It could explain the persistence of low Ni and the large Ri. For F13 flight, MYKE2

and MYK4 simulate relatively well the TIC1 formation above 450 hPa in the observation range while below 450 hPa, they, both, overestimate Ri by factor 2. For this TIC1 cloud, MYKE2 and MYKE4 give the smallest error in comparison to REF. For F21 flight, MYKE2 and MYKE4 improve the comparison of simulated Ri against observations, showing large ice crystals even if the cloud top altitude is underestimated. For F29 flight, observed values of Ri are even larger. MYKE2 and MYKE4 show a little improvement in comparison to REF, only above around 450 hPa with larger simulated ice crystals than REF. For

both F21 and F29 flights, MYKE2 and MYKE4 underestimate the observed Ri by factor 2.





## 4.4 Discussion

Our analysis shows the poor performance of the original REF parameterization in representing ice heterogeneous nucleation with low IWC and reveals that MYKE parameterization can improve significantly the representation of the IWC at all vertical levels in polluted or unpolluted air masses. Along the three flights, RHi is therefore lower in the MYKE2 and MYK4 simulations than in the REF run at cloud top. This may be due to the new parameterization promoting ice nucleation by a reduction of the available supersaturated water vapour. The new parameterization with the variation in time and space of $A_d$ and $N_0$ better represent Ni and Ri values at the top of TICs for F13 and F21 flights where the nucleation occurs. The pronounced slope of observed Ri above 500 hPa level in TIC2 cases (Fig. 8) indicates a rapid growth of the ice crystals which consume supersaturated water vapor faster than it is made available in the model. Finally, for F29 flight, the new parameterization improves slightly Ri at the top of the clouds, while; under around 450 hPa level, simulated results show better agreement for the REF simulation. The reason for that is not clear. However, Fig. 5 shows a decrease of theta with the altitude between 450 and 500 hPa in connection with an increase of ammonium molar concentration (Fig. 4B), which leads to a more efficient heterogeneous nucleation of ice at this altitude with smaller ice crystals and larger concentrations. Finally, from the comparison of the three cases simulations, we can assess the ability of the new scheme to discriminate TIC1 and TIC2 clouds. For F13, while REF results in a TIC2 cloud, MYKE2 and MYKE4 simulations produce a TIC1 in agreement with observations. As shown before, the order of magnitude of Na at the top of the cloud for F13 and F29 are similar but the f factor shows more acidic aerosols for F29. For both cases, close values of IWC allow us comparing MYKE results of Ni and Ri. Looking at the top of the cloud (above 440 hPa level), Ni, is lower for F29 than for F13 and Ri is larger for F29 than for F13, responding to acid aerosol through the variation of the contact angle. Within the limit of our calculation, the new parameterization improves significantly the representation of nucleation in TIC1 for F13 versus a TIC2 for F29 at the cloud tops, despite the model's bias of simulated aerosols by WRF-Chem over Arctic (Mölders et al., 2011). The comparison between simulations of F21 and F13 cases with MYKE is not so clear. Even if, at the top of the cloud, Ni is lower for F21 than for F13 as expected, Ri is smaller for F21 than for F13, which is not consistent with TIC types. However, the comparison of f factor at the cloud tops shows similar values for F21 and F13 near acid neutrality. This result highlights the importance of a consistent simulation of aerosol physicochemical properties to get a valuable simulation of microphysical ice cloud properties with our new parameterization of heterogeneous ice nucleation. In general, regarding overall simulated results; MYKE4 shows better agreement with observations than MYKE2 either for TIC1 or TIC2 clouds. It is well known that the effect of acid coating on IN is to reduce its ability to form ice crystal and, this effect increases with the amount of acid (Sullivan et al., 2010; Yang et al., 2011). Moreover, our results suggest that even a low acidity on IN leads to an important decrease of the heterogeneous ice nucleation rate because, for MYK4, $\theta$ increases more rapidly when acid coating increases i.e. decrease of $f$ factor (Fig. 1).

## 5 Conclusions

A new parameterization of ice heterogeneous nucleation based upon CNT approach and coupled with real time chemistry information is proposed in WRF-Chem model. The coupling with chemistry links the contact angle $\theta$ to the neutrality factor





of aerosols, which is a good proxy for the acidity of aerosols. This new parameterization is implemented in the (Milbrandt and

Yau, 2005a, b) two-moment cloud microphysical scheme available in WRF-Chem. It is particularly designed to simulate Arctic ice clouds. In the Arctic, ice clouds are separated into two classes: (1) TIC1 clouds characterized by large concentrations of very small crystals, and TIC2 clouds characterized by low concentrations of larger ice crystals. TIC2 clouds induce significant ice crystal precipitation or so-called diamond dust, a notoriously deficient variable to simulate in polar atmospheric models despite its significant contribution to the annual snow fall and generally reported as "trace" by station observations. The model

including the original Milbrandt and Yau scheme and the modified one are applied to three test cases observed during the ISDAC campaign: one TIC1 and two TIC2 clouds. For each case, results are analyzed in terms of meteorology, chemistry and cloud microphysical properties by comparison between new (MYKE2 and MYKE4) and original (REF) cloud microphysical scheme and with available observations. Our results show the poor performance of the REF parameterization in representing Arctic ice cloud types at low IWC and underline that MYKE2 and MYK4 parameterizations significantly improve the rep-

resentation of the IWC, especially in the top region of the clouds where nucleation dominates, in polluted or unpolluted air masses. MYKE2 and MYKE4 simulations is in better agreement with observation for the three flights. On the contrary, REF always strongly underestimates IWC values with a negative bias and does not see any noticeable IWC cloud at these levels on F21 flight. Aerosol number concentrations are simulated with the same order of magnitude than observations under 550 hPa level, whereas, above 550 hPa level, the simulated value is overestimated by a factor 3 for F13 flight and is underestimated

by one order magnitude for F29 flight. Despite known difficulties in simulating aerosol concentrations in WRF-Chem over the Arctic region (Mölders et al., 2011)), our parameterization achieves to represent proper cloud types, TIC1 for F13 flight versus a TIC2 for F21 and F29 flights in the nucleation region at the cloud top. Values and vertical structures of ammonium and sulphate molar aerosol concentrations for F21 and F29 flights correspond fairly well to mean observed concentrations i.e. 7 $nmol/cm^3$ and 5.5 $nmol/cm^3$ during ARCTAS and ARCPAC campaigns respectively with known contributions from volcanic

sources, peaking in the mid-troposphere. MYKE2 and MYKE4 simulations are similar showing higher $\theta$ values at clouds top for F21 and F29 flights in comparison to F13 flight, thus differencing the acidic to the nonacidic cases as expected and a low sensitivity to the arbitrarily parameterized curve shape. For the TIC1 case, REF strongly underestimates the ice crystal number concentration by at least two orders of magnitude and overestimates the mean radius, resulting in the false representation of an ice cloud, corresponding rather to a TIC2. On the contrary, the new parameterization captures well the cloud type, with repre-

sentative microphysical structure (IWC, ice crystal mean radius and ice crystal number concentration) at the top of the cloud where the nucleation occurs. TIC2 clouds observed along F21 and F29 flight tracks are characterized by a small concentration of ice crystals ranging between 1 and 30 $L^{-1}$. MYKE2 and MYKE4 simulate those ice crystal number concentrations within the ranges of observations uncertainties. For F21 flight, REF is not able to simulate a persistent cloud, while both MYKE2 and MYKE4 simulations show a cloud with ice crystal concentration close to observations. Corresponding values are typical of

TIC2 cloud under 450 hPa level, even if, the model underestimates the cloud top altitude, as the result of biases in the simulated temperature and relative humidity over ice. MYKE2 and MYKE4 also improve the ice crystal mean radius showing larger ice crystals than REF. For F29 flight, both MYKE2 and MYKE4 show an increase in ice crystal concentration compared to REF, which has the best statistics, but MYK2 and MYKE4 results are still overestimated by one order of magnitude. MYKE2 and





MYKE4 slightly improve the representation of ice crystal mean radius in comparison to REF above 450 hPa level with larger

simulated ice crystals than REF. For both TIC2 flights, MYKE2 and MYKE4 nevertheless underestimate the observed mean
radius by factor 2. Since, the Milbrandt and Yau scheme does not account for sedimentation of ice crystal, like diamond dust
type, the model consistently underestimates the ice crystal concentration in the lower cloud region. This would be improved
by adding a prognostic "diamond dust" type of hydrometeor in a future version. (same paragraph) Comparing the two versions
of the parameterization, for the three cases, in general, MYKE4 presents a slight improvement as compared to MYKE2 in

agreement with $\theta$ dependency. In our simulations, the secondary organic aerosols (SOA) formation is not considered. How-
ever, the concentration of their precursor species, mainly biogenic and aromatic volatile organic compounds, should be low
in the ISDAC campaign region and period as suggested by WRF-Chem simulation. However, results obtained later during the
NETCARE campaign (2015) shows a potential contribution of SOA to the total mass of Arctic aerosols, but their precursors
are not yet identified in the Arctic, a new challenge in simulating their formation Abbatt et al. (2019). Moreover, as our param-

eterization is dedicated to the simulation of Arctic ice cloud types, we are confident that the combination of CBM-Z-MOSAIC
is appropriate even if CBM-Z is a relatively simple gas-phase mechanism and if SOA formation is not considered. Indeed, our
results suggest that it is enough to consider the chemical impact on heterogeneous ice nucleation though the degree of aerosol
acidity acting as IN. Despite the huge challenge, our parameterization seems promising. Further studies will help validations
against satellite data and future campaigns. In particular, future flight campaigns should include simultaneously measurements

of cloud microphysics properties, of aerosols number size distribution, of aerosols chemical composition and of ice nuclei
number concentrations. The next step will be to extend simulations to quantify the role of ice nucleation of acid pollution on
radiation and atmospheric water balance, and ultimately, on the Arctic climate.

*Code and data availability.*    The code is available on request from the corresponding author

*Author contributions.*    SAK and EG devellopped and inplemented the parameterization with support of JCR and ML. SAK performed the

simulations with technical support of TO. SAK analysed results and wrote the paper with support of JCR, ML, JPB. All authers contributed
to the paper and to the analysis

*Competing interests.*    The authors declare that they have no conflict of interest

*Acknowledgements.*    As we know that Eric Girard had numerous discussions with our colleague Allan Bertram on the form of the dependency
between the contact angle and the neutrality factor, we would like to thank him for his invaluable help. We thank NETCARE (Network on

Climate and Aerosols: Addressing Key Uncertainties in Remote Canadian Environments) and NSERC (Natural Sciences and Engineering
Research Council of Canada) for funding support and ARM (Atmospheric Radiation Measurement Program) for the data collected during





ISDAC. Computer modeling benefited from access to IDRIS HPC resources (GENCI allocations A003017141 and A005017141) and the IPSL mesoscale computing center (CICLAD: Calcul Intensif pour le CLimat, l'Atmosphère et la Dynamique).



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





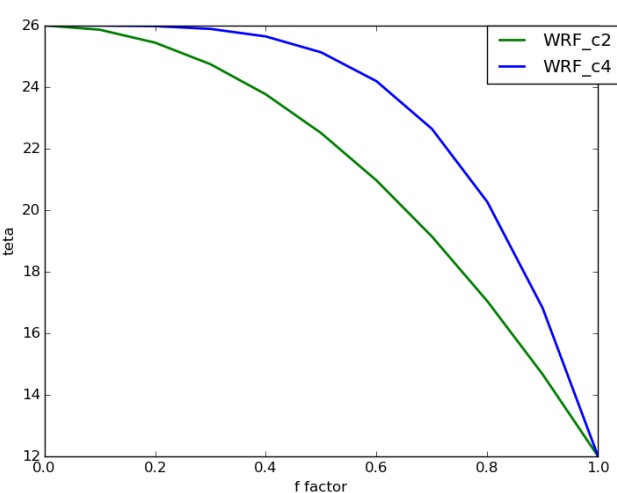

**Figure 1.** Variation of $f$ with ($\theta$) for MYKE2 (blue line) and MYKE4 (green line).



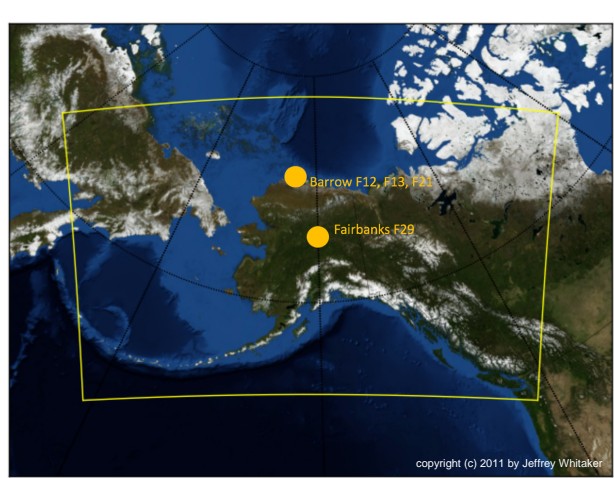

**Figure 2.** Model domain (yellow) used in this study centred over Fairbanks with a horizontal resolution of 10 km. The cities of Barrow (71.18,-156.44) and Fairbanks (64.83,-147.77) where F12, F13, F21 and F29 flights are based are also shown with orange dots.





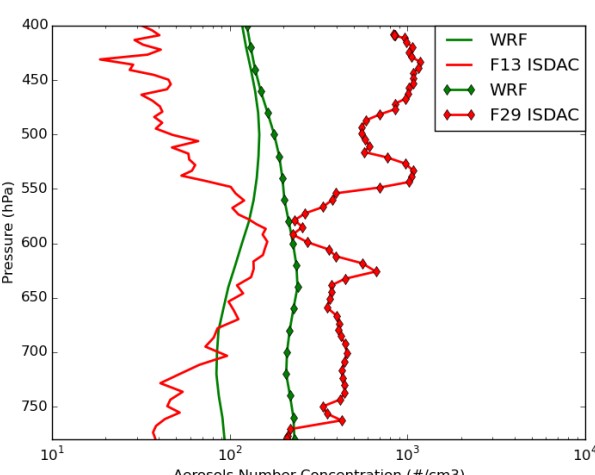

**Figure 3.** Comparison of the observed (red) and simulated (green) WRF vertical profiles of aerosol number concentrations. Observations were measured by the PCASP in situ sensor on board the Convair-580 just before entering the clouds for F13 (solid lines) and F29 (solid lines with diamond markers) flights. Note that PCASP measurements were not available during F21 flight.





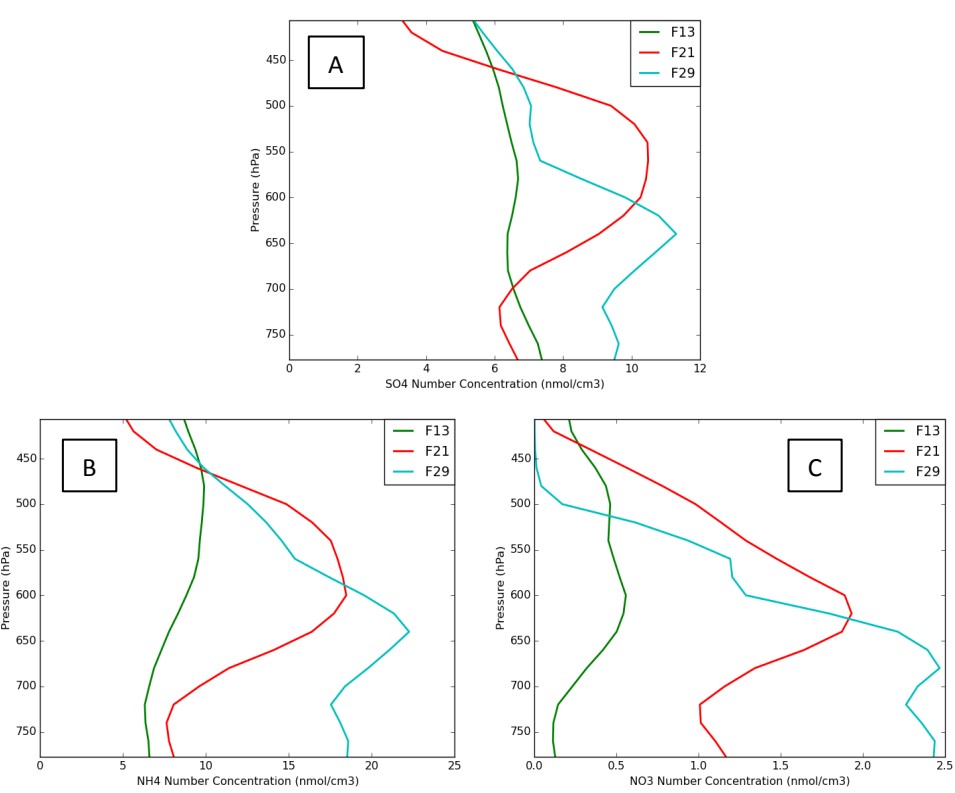

**Figure 4.** Vertical profiles of sulphate (A), ammonium (B) and nitrate (C) molar aerosol concentration along F13 (green), F21 (red) and F29 (light blue) flights.





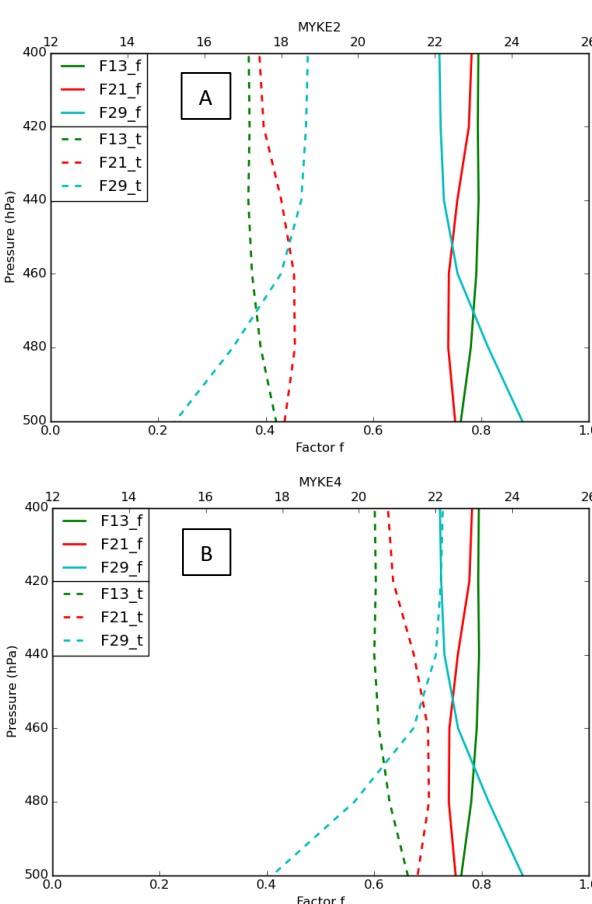

**Figure 5.** Vertical profiles of the factor $f$ (full line) and the contact angle ($\theta$) (dashed) for MYKE2 (A) and MYKE4 (B) along F13 (green), F21 (red) and F29 (light blue) flights.





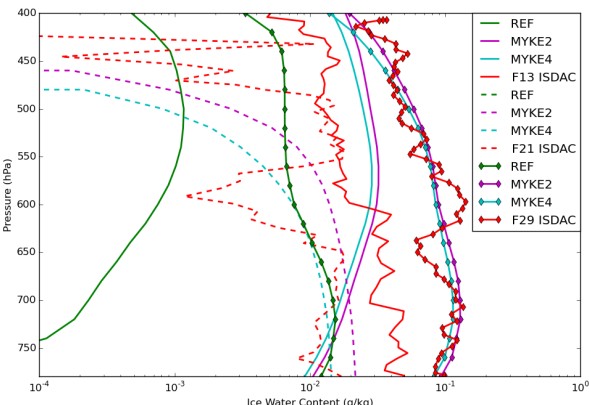

**Figure 6.** Comparison of the observed (red) and simulated (REF in green, MYKE2 in purple and MYKE4 in cyan) vertical profiles of IWC along F13 (solid lines), F21 (dashed lines) and F29 (solid line with diamond markers) flights.



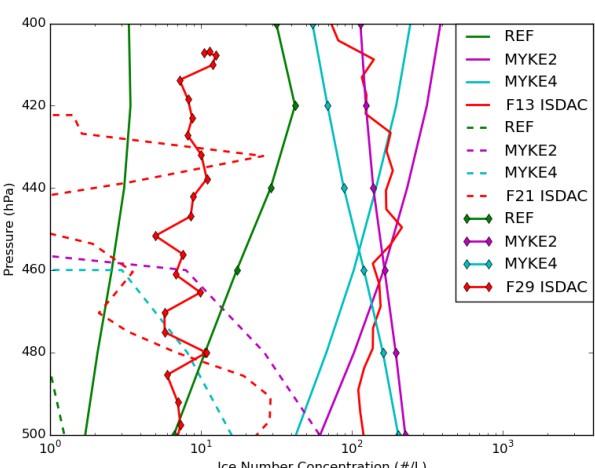

**Figure 7.** Comparison of the observed (red) and simulated (REF in green, MYKE2 in purple and MYKE4 in cyan) vertical profiles of Ni along F13 (solid lines), F21 (dashed lines) and F29 (solid line with diamond markers) flights.



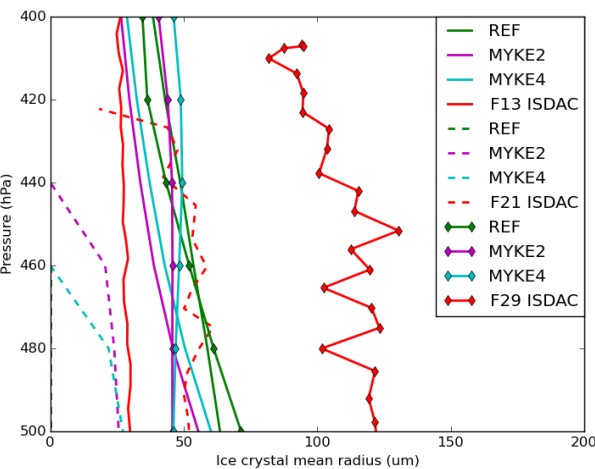

**Figure 8.** Comparison of the observed (red) and simulated (REF in green, MYKE2 in purple and MYKE4 in cyan) Ri along F13 (solid lines), F21 (dashed lines) and F29 (solid line with diamond markers) flights.





**Table 1.** Bulk density for each hydrometeor category

| Hydrometeor category | Hydrometeor Bulk density (kg/m3) |
|---|---|
| Cloud | 1000 |
| Rain | 1000 |
| Cloud ice | 500 |
| Snow | 100 |
| Graupel | 400 |
| Hail | 900 |





**Table 2.** Source and sink terms, listed according to the hydrometeor category, which gains mass/number, except for self-collections or when the lost is to water vapor.

| Hydrometeor | Source terms |
|---|---|
| Cloud | nucleation, condensation/evaporation, self-collection |
| Rain | autoconversion, evaporation, accretion of cloud, self-collection, melting of frozen hydrometeors |
| Ice nucleation | (contact, deposition, condensation-freezing, rime splintering, immersion, homogenous freezing of cloud), riming of cloud, deposition/sublimation |
| Snow | conversion from ice (including ice aggregation), collection of ice and cloud, deposition/sublimation, aggregation (self-collection), collisional freezing with rain |
| Graupel | collisional freezing of rain and ice/snow/graupel, conversions from ice and snow, collection of cloud and ice, deposition/sublimation |
| Hail | collisional freezing of rain and ice/snow/graupel, collection of cloud/rain/ice/snow, deposition/sublimation, probabilistic freezing of rain, conversion from graupel |



**Table 3.** Parameterizations and options used for the WRF-CHEM simulations.

| Meteorological option | Selected option |
| --- | --- |
| Microphysics | (Milbrandt and Yau, 2005) |
| SW radiation | RRTMG (Iacono et al., 2008) |
| LW radiation | RRTMG (Iacono et al., 2008) |
| Cumulus parameterization | KF-CuP (Berg et al., 2015) |
| Planetary boundary layer | MYJ (Janjic, 1994) |
| Surface layer | Monin-Obukhov Janjic Eta scheme (Janjic, 1994) |
| Land surface | Unified Noah land-surface model (Chen and DUDHIA, 2001) |
| Chemistry and aerosols options | |
| Gas-phase chemistry | CMB-Z (Zaveri et al., 2008) |
| Aerosols | MOSAIC 8-bins (Zaveri et al., 2008) + VBS-2 SOA formation and aqueous chemistry |
| Photolysis | Fast-J (Wild et al., 2000) |





**Table 4.** Root mean square errors (RMSE), biases (Bias) and correlation coefficients (Cor) or the temperature (T) and relative humidity over ice (RHi) for the three simulations (REF, MYK2 and MYKE4).

| Flight | Variable | Simulation | RMSE | Bias | Cor |
|---|---|---|---|---|---|
| | | REF | 1.92 | -1.90 | 0.99 |
| | T | MYKE2 | 1.76 | -1.72 | 0.99 |
| | | MYKE4 | 1.77 | 1.73 | 0.99 |
| F13 | | REF | 10.86 | 8.55 | 0.95 |
| | RHi | MYKE2 | 17.74 | 15.58 | -0.61 |
| | | MYKE4 | 17.08 | 14.88 | -0.26 |
| | | REF | 3.30 | -3.00 | 0.82 |
| | T | MYKE2 | 3.31 | 3.02 | 0.82 |
| | | MYKE4 | 3.30 | 3.01 | 0.82 |
| F21 | | REF | 55.71 | 51.68 | -0.06 |
| | RHi | MYKE2 | 56.02 | 52.28 | -0.03 |
| | | MYKE4 | 55.84 | 51.93 | -0.05 |
| | | REF | 2.65 | 2.64 | 0.99 |
| | T | MYKE2 | 2.17 | 2.16 | 0.99 |
| | | MYKE4 | 2.19 | 2.18 | 0.99 |
| F21 | | REF | 11.86 | 11.37 | 0.67 |
| | RHi | MYKE2 | 16.67 | 16.31 | 0.65 |
| | | MYKE4 | 16.12 | 15.79 | 0.69 |