# Peer review of "A new parameterization of ice heterogeneous nucleation coupled to aerosol chemistry in WRF-Chem model version 3.5.1: evaluation through the ISDAC measurements"

_Geoscientific Model Development, 2020_

## Referee Comment (RC1) · Anonymous Referee #1 · 4 May 2020

The authors present a new parameterization for ice crystals formation by heterogeneous ice nucleation coupled to aerosols chemistry in the WRF-Chem model. They conclude that "the new parameterization is, thus able to represent TIC1 and TIC2 microphysical characteristics at the top of the clouds where heterogeneous ice nucleation is most likely occurring even knowing the bias of simulated aerosols by WRF-Chem over Arctic." This paper could represent a valuable addition how to better represent ice formation in Arctic clouds. But I am sad to say that the paper is currently in a poor shape as far as its presentation is concerned. I also have some major comments about

the new parameterization and some of the results. However, if the following comments are thoroughly addressed within this review process I would suggest publishing this paper in GMD.

**1 Major comments**

The first major comments concern the condition of the paper presentation. Usually this would be part of the minor comments section. However, several presentation errors popped up while reading through the paper, which is the reason why it is already mentioned here. My impression is that the paper was not checked at the end for typos, bracket errors, consistency of parameter notation/writing style (regular vs. italic), description of abbreviations, etc. I will give some examples:

Abstract, line 1, 6 and 10: What do the abbreviations TIC, WRF-Chem and IWC stand for?
Line 25: It should read "... understand fundamental processes...".
Line 26: Delete one "are" in "...which are are particularly...".
Line 239 and 240: There are no Eqs. (20a) and (20b), only Eqs. (20) and (21).
Line 273: There is no Table 5. It should be Table 4.
Line 352: There is a "10-3" too much between -8.1 $10^{-3}$ g/kg.
There are more of such errors. Please check the paper thoroughly.

If you use a citation as a constituent part of the sentence (e.g. grammatical subject) then please check your brackets. For example, line 52: It should read "In Keita et al. (2019), the parameterization of Girard et al. (2013) based upon CNT..."

Some of the parameters used are not / differently introduced or differently written, etc. Some examples:

$N_i$ vs Ni, $S_i$ vs Si, $f$ vs f, U vs $U$, etc.

'Ri' is not introduced (from the context it is ice particle radius), but in the model description you use diameter.

Line 101 -103: Please avoid the brackets around '$\alpha_x$', '$\lambda_x$', etc.

Please also check your equations. For example Eq. (2): There is the wrong 'dot' in the scalar product. And what does '*TURB*' stand for or in other words how does this term including '*TURB*' look like?

The second major comments concern the new parameterization and some of the results.

You mention on page 6, starting line 151: "The new parameterization focuses on deposition ice nucleation for uncoated IN and to immersion freezing of sulphuric acid coated IN, i.e. IN immerged in an acid aqueous solution." Here I wonder that for $\Delta G$ in the presentation of the nucleation theory you only consider the case of deposition nucleation. If you look into the literature, e.g., Lamb and Verlinde (2010, pages 313-318) you can see that there is a difference between the Gibbs free energies between the case of deposition nucleation and immersion freezing due to e.g. the differences in the interfacial free energies. Finally, you would end up with different nucleation rates even if identical contact angles would be used in the nucleation rate equation. What about the freezing point depression when the particle is immersed in an aqueous solution? Could you please comment on that?

Concerning the parameterization of the contact angle: Why do you use the quadratic and biquadratic forms? Eastwood et al. (2008) show the ice nucleation behavior of various minerals and the respective contact angles. Why did you choose the contact angle of kaolinite? Note that Eastwood et al. (2008) used a different (simplified) equation of the reduction factor in contrast to your Eq. (17). What are the consequences when using contact angles based on Eastwood et al.? Have you also considered checking other papers for contact angles? For my impression the contact angles given

in Eastwood et al. are smaller compared to other studies of kaolinite (e.g. Welti et al. (2012), with $\theta \approx 90°$ for kaolinite particles in the immersion freezing mode)

Concerning the representation of the IWC by MYKE2 and MYKE4: Looking on figure 6, the IWC for F29 is well reproduced by MYKE2 and MYKE4. However, looking on Figs. 7 and 8, which show the vertical distribution of the ice particle number concentration and the ice particle radius (the combination of both at the end leads to IWC), you can see that MYKE2 and MYKE4 overestimate ice particle number concentration and underestimate ice particle radius for F29. Putting these two factors now together lead to a good IWC, however, to my impression just by chance. Actually, for the vertical distribution of the ice particle number concentration alone, REF does a better job. In my view, MYKE2 and MYKE4 are not able to correctly represent the TIC2 microphysical characteristics. Could you please comment on that?

**2  Minor comments**

Following the recommendation of Vali et al. (2015), I would suggest to use "ice nucleating particles (INPs)" instead of "ice nuclei (IN)".

The year in the citation "Pruppacher and Klett (1998)" should either be 1997 or 2010. 1998 refers to a review of that book.

Line 95 and Eq. (4): Why is this equation explained and written in such a complicated way? For me it looks like to simply be density times volume for the hydrometeors considered: $m_x(D) = \rho_x V_x = \pi/6 \rho_x D^3$

[Figure]

Line 124 -124: You only consider homogeneous ice nucleation of pure supercooled water droplets. What about haze droplets and the resulting freezing point depression?

Eqs. (20) and (21) and Fig.1: Could you please make clear in the text, when introducing Eqs. (20) and (21), that Eq. (20) belongs to MYKE2 and Eq. (21) to MYKE4? It is mentioned in Fig. 1 but not in the text at the end of section 2.1.1.

**3  References**

Lamb, D. and Verlinde, J.: Physics and chemistry of clouds, Cambridge University Press, Cambridge, UK, 2011.

Vali, G., DeMott, P. J., Möhler, O., and Whale, T. F.: Technical Note: A proposal for ice nucleation terminology, Atmos.  Chem.  Phys., 15, 263–270, https://doi.org/10.5194/acp-15-10263-2015, 2015.

Welti, A., Lüönd, F., Kanji, Z. A., Stetzer, O., and Lohmann, U.: Time dependence of immersion freezing: an experimental study on size selected kaolinite particles, Atmos. Chem. Phys., 12, 9893–9907, https://doi.org/10.5194/acp-12-9893-2012, 2012.

---

## Short Comment (SC1) · 17 Jul 2020

This is an executive editor comment highlighting the ways in which this manuscript is not currently compliant with GMD policy on code and data availability. In this case, no code or data has been published at all. This is completely incompatible with the GMD code and data policy, and unless remedied swiftly must result in the rejection of the manuscript.

GMD requires that the model code, input data, configuration files, run and analysis

scripts be publicly archived before the manuscript is submitted. It is unfortunate that this manuscript has got as far as GMDD without this being remedied, however swift action is now required.

Further details on code and data availability requirements are in the GMD model code and data policy: https://www.geoscientific-model-development.net/about/code_and_data_policy.html. The reasons for the policy and more detail are provided in this editorial: https://doi.org/10.5194/gmd-12-2215-2019. Please immediately archive the required data and post the citations in a response to this comment. This will enable any reader who wishes to examine the data on which the paper is based to do so as part of the public review process. The code and data availability section of the revised manuscript must then properly cite the archived code and data.

---

## Referee Comment (RC2) · Anonymous Referee #2 · 20 Jul 2020

In this work the authors develop a new parameterization of heterogeneous ice nucleation targeted specifically to Arctic clouds. The authors use classical nucleation theory, CNT, to estimate ice nucleation rates. A novel aspect of this work is the usage of the degree of neutralization of the aerosol to parameterize the contact angle used in CNT. The authors apply the parameterization in the simulation of two cloud formation cases during the Indirect and Semi-Direct Aerosol Campaign (ISDAC). Compared to the reference parameterization, the new approach leads to a better agreement between the observed and simulated ice crystal number concentration. This is an interesting work,

of interest to the atmospheric community. The authors should expand on the rationale behind the proposed parameterizations. The presentation needs some work. After these issues have been addressed, that manuscript would be suitable for publication.

General comments.

My main concern is related to the lack of a proper justification for the proposed parameterizations. The authors base their development on CNT which accuracy for heterogeneous ice nucleation is still matter of debate, although it has been applied before. However the authors make some assumptions that need to be justified. Ice nucleation is assumed to occur mainly in the deposition mode or by immersion in solution. As mentioned by another reviewer only expressions for deposition ice nucleation are used. Moreover, why are these considered the main paths of ice nucleation in the stratiform clouds? Droplet freezing is probably more significant. If not, the authors should show some evidence or at least reports suggesting otherwise. Also, a control simulation where CNT is used but with no acidity dependency considered should be added to discriminate the effect of the later.

Minor comments.

Lines 16-19. Please split this sentence.

Line 26. Should be "specific"'.

Line 29. Remove the comma.

Line 33. Number density is however a function of temperature.

Line 34. CNT is not a requirement of the stochastic hypothesis. Please rephrase.

Lines 36-39. Most atmospheric models use time-independent formulations. In fact, all of these references correspond to time-independent formulations.

Line 41. Please explain the significance of the contact angle. Also isn't this the approach used in this work? A single contact angle, dependent on the acidity?

Line 45. Say INP (ice nucleating particle) instead of IN.

Line 55. Is dust internally mixed with sulfuric acid?

Line 103. Is this assumption appropriate for small ice particles?

Line 133. Why is immersion freezing of cloud droplets (which is likely the dominant path of ice formation) not treated in a more rigorous way?

Line 159, Eq. 13. Is this the total surface area? Shouldn't this equation be weighted by the aerosol size distribution? Also, when applying this to the immersion case, shouldn't it be only valid for the dust particles immersed within the haze aerosol droplets?

Line 170. This seems wrong. Is it maybe 10ˆ26?

Line 174. The surface tension between ice and vapor is a function of temperature. Also, this would be invalid for immersion within haze particles.

Line 176. This is not the expression for an infinite plane surface. This is in fact the expression for small INP when the size is comparable to the size of the ice germ.

Line 203, Eq. 19. Is this for the dust particles internally mixed with sulfate and nitrate, or the overall composition? The latter would not seem very rigorous. Please explain.

Line 215-220. What is the rationale behind the proposed functional forms in Eqs. 20 and 21? Why would the contact angle depend on the acidity?

Line 233. There are no equations 20a and 20b.

Line 265. If ice nucleation occurs at cloud top why would it be on haze aerosol instead of cloud droplets immersed with dust?

Line 285. Is this the total aerosol number for all species?

Line 327. This is a confusing sentence? What do the authors mean by the same f?

Line 349-350. Can you show this in a plot?

Lines 384-385. Please show this.

Line 402-403. What about using no f, i.e., Just a fixed contact angle?
* * *

---

## Referee Comment (RC3) · Anonymous Referee #3 · 31 Jul 2020

This paper by Keita et al. introduces a new parameterization in one of the microphysical scheme of the WRF-Chem model to better represent the impact of the degree of acidity in clouds on the nucleation rate and cloud microphysical properties. To validate this method, it uses in-situ measurements in Arctic clouds made by an aircraft, which is always a difficult thing to do. This is a challenging goal, which must be addressed with thoroughness. I'm afraid it is not the case in the present version of the paper. The presentation is poor (lack of clarity, long paragraphs) and there is a lack a proofreading (mistakes in equations, typos). But more importantly, the authors do not discuss the

origin of their parameterization, which is one possibility among others, in an already complex scheme. The results are encouraging but should be described in more detail. I have read really good papers by the same team and they are recognized experts in the field. For some reason, the submitted version of this paper is too preliminary. This paper is definitely useful and contains interesting ideas. It will be of interest to the community, but I would clearly suggest a rewrite by the authors, as well as a second wave of peer review, before publication.

Major comments

First major comment: the rational behind this new parameterization is not clearly presented. Section 2.1.1 (which should be section 2.2) must be rewritten. Why did the authors decide to change the nucleation rate and the contact angle, instead of another method ? Why did they choose this relationship between the neutralized fraction and the contact angle ? How does this new parameterization fit in the Milbrandt and Yau scheme exactly (a diagram would help) ? This section is confusing and incomplete.

Second major comment: The paper clearly lacks proofreading. A lot of well-known and well-established equations contain mistakes.

For example, these equations contain mistakes, and there might be other mistakes that I missed :

Equation 1: velocity is missing in the first term of the right-hand side part of the equation, a dot is missing as well (convergence) ; the third term is d/dz and not d/dt

Equation 4: it is a PDF, therefore, Nx(D) = dn/dD, and writing dNx(D) does not make any sense. Ntx is the total number concentration, and is integrated over D, so it is Ntx and not Ntx(D). In the exponential, both lambda_x and D are to the power of nu_x, not only D, it is therefore (lambda_x D)^nu_x.

Equation 5: again, it is a PDF, and it is Nx(D) and not dNx(D).

Equation 6: Ntx and not Ntx(D)

[Figure]

Equation 8 is not consistent I believe; it is not in kg/kg, because of the 1/rho factor.

Equation 11: I don't understand where this equation comes from. Please demonstrate.

Equation 16: usually Mŵ2 also appears in the Gibbs free energy term;

Equation 17: it is not q-qcos(theta) but q-cos(theta)

Minor comments :

l.85: "All symbols for variables and parameters used are listed in Table 1." Where is Table 1 ? It appears to be missing. This probably explains why the numbering of all the other tables is wrong...

l.155: "For condensation-freezing, it can be included in the immersion freezing of coated IN when air is supersaturated with respect to liquid water." This sentence is quite confusing, and this whole paragraph is unclear. How does this new parameterization fit in the Milbrandt and Yau scheme exactly ? Please include a diagram, for example.

Sections 4.2, 4.3, 4.4, 5: these sections are all made of one huge paragraph and are very hard to read.

l.606: the two references to Milbrandt and Yau are the same, and should be Part I and part II;

Figure 6 is very hard to read.

–––––––––––––––––––––––––––––––

---

## Author Comment (AC1) · 5 Sep 2020

The authors thank Reviewer#1 for this comprehensive review of the paper. We address below each comment individually (in blue color). Line numbers refer to the original manuscript.

The first major comments concern the condition of the paper presentation. Usually this would be part of the minor comments section. However, several presentation errors popped up while reading through the paper, which is the reason why it is already mentioned here. My impression is that the paper was not checked at the end for typos, bracket errors, consistency of parameter notation/writing style (regular vs. italic), description of abbreviations, etc.
The authors apologize for those typos and errors in the style. We have carefully corrected all of them.

I will give some examples: Abstract, line 1, 6 and 10: What do the abbreviations TIC, WRF-Chem and IWC stand for?
Those abbreviations had been originally detailed in the text of the paper but, as pointed by Reviewer#1, we have forgotten to repeat them in the Abstract. They have now been clarified in the Abstract as well.

Line 25: It should read ". . . understand fundamental processes. . .".
Done.

Line 26: Delete one "are" in ". . .which are are particularly. . .".
Done.

Line 239 and 240: There are no Eqs. (20a) and (20b), only Eqs. (20) and (21).
Done. These equations become Eq. 19 and Eq. 20 in the revised manuscript.

Line 273: There is no Table 5. It should be Table 4.
Done.

Line 352: There is a "10-3" too much between -8.1 $10^{-3}$ g/kg. There are more of such errors. Please check the paper thoroughly.
We checked the entire paper for such errors and corrected all of them.

If you use a citation as a constituent part of the sentence (e.g. grammatical subject) then please check your brackets. For example, line 52: It should read "In Keita et al. (2019), the parameterization of Girard et al. (2013) based upon CNT. . ."
Done, we also checked all citations.

Some of the parameters used are not / differently introduced or differently written, etc. Some examples:
$N_i$ vs Ni, $S_i$ vs Si, $f$ vs f, $U$ vs U, etc. 'Ri' is not introduced (from the context it is ice particle radius), but in the model description you use diameter.
Physical quantities and variables are now typeset in italic font, as indicated by the recommendations of GMD journal. Abbreviations from 2 letters are typeset in roman font (e.g. $Rh_i$). Vectors are identified in bold italic font and matrices in bold roman font.

Line 145, $N_i$ has been changed to $N_{m,i}$. $R_i$ stands for the mean ice crystal radius.
Line 368 we have added "the mean ice crystal radius ($R_i$)".
Line 201 has been rewritten as: " The coupling is done by expressing $\theta$ as a function of the aerosol neutralization fraction $f_n$ in dust particles internally mixed with sulphate, nitrate and ammonium (Zhang et al., 2007; Fisher et al., 2011), which is between 0 and 1 and is defined as:"

Line 101 -103: Please avoid the brackets around '$\alpha x$', '$\lambda x$', etc.
We suppressed the brackets around all variables in the manuscript.

Please also check your equations. For example Eq. (2): There is the wrong 'dot' in the scalar product. And what does 'TURB' stand for or in other words how does this term including 'TURB' look like?

We disagree with Reviewer#1 as Eq. (2) includes a divergence. We have also added a dot to Eq. (1) according to Milbrandt and Yau (2005a) and Ferrier (1994).

We checked all the equations. From Eq. (4) and (6) we removed $(D)$ from $N_{Tx}(D)$. From Eq. (5) and (4), $dN_x(D)$ has been replaced by $N_x(D)$.

According to Milbrandt and Yau (2005a), Ferrier (1994) and Khvorostyanov and Curry (2014, pages 171-172), the TURB term is:

$$TURB = \frac{\partial}{\partial x} k_x \frac{\partial}{\partial x} + \frac{\partial}{\partial y} k_y \frac{\partial}{\partial y} + \frac{\partial}{\partial z} k_z \frac{\partial}{\partial z}$$

where $k_x$, $k_y$ and $k_z$ are the components of the turbulent exchange coefficient. Eq. 1 and Eq. 2 have therefore been rewritten to include the turbulent diffusion matrix.

The second major comments concern the new parameterization and some of the results. You mention on page 6, starting line 151: "The new parameterization focuses on deposition ice nucleation for uncoated IN and to immersion freezing of sulphuric acid coated IN, i.e. IN immerged in an acid aqueous solution." Here I wonder that for ΔG in the presentation of the nucleation theory you only consider the case of deposition nucleation. If you look into the literature, e.g., Lamb and Verlinde (2010, pages 313-318) you can see that there is a difference between the Gibbs free energies between the case of deposition nucleation and immersion freezing due to e.g. the differences in the interfacial free energies. Finally, you would end up with different nucleation rates even if identical contact angles would be used in the nucleation rate equation. What about the freezing point depression when the particle is immersed in an aqueous solution? Could you please comment on that?

The authors thank Reviewer#1 for these comments.

Our objective in developing the new parameterization was to represent the formation of ice crystals in the particular conditions of Arctic TIC clouds. In these conditions, it is mainly the deposition mode that occurs for the heterogeneous nucleation of ice, i.e. the air mass is in water-subsaturated regime. Kulkarni et al. (2014) showed that, except for quartz, acid-coated dusts are less effective INPs in the deposition mode but have similar effectiveness in the immersion-freezing mode, i.e. in water-supersaturated regime. Based on X-ray diffraction analyses, they argued that acid treatment caused structural deformations of the surface dusts, and the lack of structured order reduced the ice nucleation properties of coated particles in the deposition mode. Moreover, they suggested that, at water-supersaturated conditions, surface chemical reactions might not change the original ice nucleating properties permanently because coating material could be removed by dissolution. For kaolinite, Panda et al. (2010) concluded that sulfuric acid-treated particles could result in the formation of aluminum sulfate that can be easily dissolved in water. Considering these recent findings, and our objective to develop a simplified parameterization to limit computational time, we chose to use the CNT formula for deposition mode but with a specific factor, the neutralization fraction, indicating the degree of acidity of the coating of dust particles. Several passages of the text have been modified to clarify the conditions of the parameterization:

Line 55: "In Keita et al. (2019), the parameterization of Girard et al. (2013) for water-subsaturated conditions based upon CNT approach was implemented in the online Weather Research and Forecasting model coupled with chemistry (WRF-Chem) (Grell et al., 2005). This parameterization is suitable to represent the formation of ice clouds in Arctic. It assumes that INPs are mainly mineral dust particles, which is consistent with recent results from the NETCARE (Network on Climate and Aerosols: Addressing Key Uncertainties in Remote Canadian Environments) project (Abbatt et al., 2019)."

Line 66: "In this paper, we investigate for the first time the ice heterogeneous nucleation in a fully coupled aerosol and chemistry parameterization."

Line 78: "The new scheme for ice crystals formation by heterogeneous nucleation in the deposition mode is implemented…"

Line 153: "Moreover, the condensation-freezing mode, as discussed in Vali et al. (2015), is quite uncertain."

Line 157: "The new parameterization focuses on the heterogeneous ice nucleation for uncoated INPs and for sulphuric acid coated INPs in the deposition mode i.e. in water-subsaturated conditions. In this approach, INPs are assumed to be mineral dust particles following Girard et al. (2013). For contact freezing and immersion freezing from supercooled cloud droplets, the parameterizations remain unchanged. As condensation-freezing is uncertain (Vali et al., 2015), this process is not longer included in the model."

Line 207: "For instance, Kulkarni et al. (2014) showed that, except for quartz, acid-coated dusts are less effective INPs in the deposition mode but have similar effectiveness in the immersion-freezing mode, i.e. in water-supersaturated regime. Based on X-ray diffraction analyses, they argued that acid treatment caused structural deformations of the surface dusts, and the lack of structured order reduced the ice nucleation properties of coated particles in the deposition mode. Moreover, they suggested that, at water-supersaturated conditions, surface chemical reactions might not change the original ice nucleating properties permanently because coating material could be removed by dissolution. Panda et al. (2010) concluded that sulfuric acid-treated kaolinite particles could result in the formation of aluminum sulfate that can be easily dissolved in water. Considering these recent findings, and our objective to develop a simplified parameterization to limit computational time, we choose to use the CNT formula for deposition mode but with a specific factor, the neutralization fraction $f_n$, indicating the degree of acidity of the coating of dust particles."

Please, note that the immersion freezing of raindrops and cloud water droplets still follows the parameterization of (Bigg, 1953) but is not activated due to the absence of liquid drops in the simulated TIC clouds, except for some few exceptions in the lower part of clouds.

Finally, the term "pre-exponential factor" at line166 has been replaced with "kinetic coefficient" in coherence with Fletcher (1958).

Concerning the parameterization of the contact angle: Why do you use the quadratic and biquadratic forms? Eastwood et al. (2008) show the ice nucleation behavior of various minerals and the respective contact angles. Why did you choose the contact angle of kaolinite? Note that Eastwood et al. (2008) used a different (simplified) equation of the reduction factor in contrast to your Eq. (17). What are the consequences when using contact angles based on Eastwood et al.? Have you also considered checking other papers for contact angles? For my impression the contact angles given in Eastwood et al. are smaller compared to other studies of kaolinite (e.g. Welti et al. (2012), with θ ≈ 90∘ for kaolinite particles in the immersion freezing mode)

The authors thank Reviewer#1 for these comments.

Keita and Girard (2016), after analysing the slope between the nucleation rate and the saturation over ice for TIC1 and TIC2 clouds (cf. Fig. 16 in Keita and Girard (2016)) observed for a given $S_i$ that: (1) the slope is the largest for the smallest accessible contact angles; (2) the decrease of the slope with the increasing contact angle is very non-linear. These results are consistent with laboratory experiments (Sullivan et al., 2010) showing a rapid increase of the contact angle with acidity on coated INP. These results motivated us to parameterize the contact angle θ as a function of the aerosol neutralization fraction under a concave form. Simple concave functions follow power law: $\theta = 26 - 14 \times f_n^p$ with p larger than 1. We have chosen a quadratic (p=2, MYKE2 simulation) form for simplicity. We have besides added a sensitivity simulation (MYKE4) under a biquadratic form (p=4) for simplicity to test the influence of the exponent p on the concave form of the contact angle with the neutralization fraction.

Kaolinite represents a significant component of mineral dust (Glaccum and Prospero, 1980). It is also found to be efficient ice nuclei in the deposition mode, requiring relative humidity with respect to ice (RH$_i$) below 112% in order to initiate ice crystal formation (Eastwood et al. 2009). This is a typical microphysical condition found in Arctic ice clouds. Recent studies from Kumar et al. (2018; 2019a; 2019b) showed that: (1) the relevance of quartz particles as atmospheric INPs is uncertain; (2) INP activity of dust particles not only depends on their composition but also on their chemical exposure history; (3) the exposition of dust particles to acidic air masses decreases their INP activity. Thus, using kaolinite as a proxy of dust particles in our parameterization is reasonable in the current state of knowledge on dust particles composition in the atmosphere, and in particular in the Arctic atmosphere where our parameterization applies.

All this discussion has been added into the revised version of the paper.

The simplified form of the reduction factor used in Eastwood et al. (2008) is appropriate for their experimental conditions where the radius of the INP is larger than the radius of the ice embryo. Unlike previous studies using the CNT approach (Keita and Girard, 2016; Keita et al., 2019; Girard et al., 2013; Khvorostyanov and Curry, 2009; Morrison et al. 2005; Liu et al., 2007; Hoose et al., 2010; Chen et al., 2008), the INP radius varies within the aerosol module in our parameterization. As a consequence, we choose the general form of the reduction factor from Fletcher (1958) including the effect of the curvature of the INP. As dust particles are mostly in the accumulation and coarse modes of the aerosol size distribution, using the simplified form of the reduction factor in our parameterization might only show small discrepancies in the results. Moreover, a typo error found in Eq. (17) is now corrected as:

"The function $f(cos\theta)$ is a decreasing function of the cosine of the contact angle $\theta$ as defined by Pruppacher and Klett (1997) for a curved substrate:

$$f(cos\theta) = \frac{1}{2}\left\{1 + \left(\frac{1-qcos\theta}{\emptyset}\right)^3 + q^3\left[2 - 3\left(\frac{q-cos\theta}{\emptyset}\right) + \left(\frac{q-cos\theta}{\emptyset}\right)^3 + 3q^2cos\theta\left(\frac{q-cos\theta}{\emptyset} - 1\right)\right]\right\},$$

(17)"

Most studies giving values of the contact angle for kaolinite focused on the immersion-freezing mode, i.e. under water-supersaturated regime. This is the case for Welti et al. (2012) for instance. This is why we consider that values of the contact angle found in these studies are irrelevant for our parameterization that concerns the water-subsaturated regime.

Concerning the representation of the IWC by MYKE2 and MYKE4: Looking on figure 6, the IWC for F29 is well reproduced by MYKE2 and MYKE4. However, looking on Figs. 7 and 8, which show the vertical distribution of the ice particle number concentration and the ice particle radius (the combination of both at the end leads to IWC), you can see that MYKE2 and MYKE4 overestimate ice particle number concentration and underestimate ice particle radius for F29. Putting these two factors now together lead to a good IWC, however, to my impression just by chance. Actually, for the vertical distribution of the ice particle number concentration alone, REF does a better job. In my view, MYKE2 and MYKE4 are not able to correctly represent the TIC2 microphysical characteristics. Could you please comment on that?

We thank the reviewer for this comment.

For F29 case, no liquid droplets are present inside the simulated cloud for both REF and MYKE. ISDAC observations showed very low liquid water content not exceeding $10^{-3}$ g/kg with a mean value around $10^{-4}$ g/kg. Such value cannot explain the observed IWC shown on Fig. 6. Among ice phase microphysical processes, the IWC is determined mainly by the ice nucleation and the solid condensation in a pure ice-phase cloud. The only difference between

REF and MYKE is the parameterization of heterogeneous nucleation of ice by deposition. In both schemes, $N_i$ is first computed and the IWC is deduced assuming the same mass of nucleated ice crystal. As in MYKE, $N_i$ is greater than in REF, the IWC is greater too with values in the same order of magnitude than ISDAC observations. We can deduce that, for F29, MYKE fails to simulate a correct cloud with an overestimation of $N_i$ and an underestimation of $R_i$ and that, even if MYKE could simulate proper value for $N_i$ and $R_i$, then IWC would be underestimated in comparison with observations. However, the F29 case seems particular in comparison to others TIC clouds observed during ISDAC. In Jouan et al. (2012), where it was analysed based on flight track above Barrow (instead of Fairbanks in Keita et al., 2019 and the present study), it was classified as a TIC1 cloud ($N_i > 10$ L$^{-1}$) with mean $N_i$ of 33 L$^{-1}$. For F29, Keita et al. (2019) showed a great difference between $N_i$ observed from ISDAC over Fairbanks and $N_i$ deduced from DARDAR observations in the upper part of the cloud (cf. Fig. 13 above 500hPa) whereas it was not the case for F21. Thus it is not clear considering results from Jouan et al. (2012) and Keita et al. (2019) if the cloud corresponding to F29 flight is a TIC1 or a TIC2. Moreover, the order of magnitude of simulated $N_i$ with MYKE for F29 is comparable to $N_i$ deduced from DARDAR.

The discussion of results for F29 lines 365-368 have been rewritten considering the above discussion:

"However, it is reasonably close to satellite observations as analysed by (Keita et al., 2019). Their analysis revealed a large discrepancy of $N_i$ between ISDAC flights and satellite estimations for F29 in the upper part of the cloud. We can notice here that the order of magnitude of $N_i$ for F29 estimated from satellite can question the classification of F29 as a TIC2 especially as Jouan et al. (2012), using flight track above Barrow instead of Fairbanks, classified this cloud as a TIC1. This discrepancy between airborne measurements, simulated results and satellite observations can be due to the small sampling domain taken during ISDAC versus the low resolution of satellite products and of the model grid."

Following the recommendation of Vali et al. (2015), I would suggest to use "ice nucleating particles (INPs)" instead of "ice nuclei (IN)".
We agree with the reviewer and we now use INP instead of IN.

The year in the citation "Pruppacher and Klett (1998)" should either be 1997 or 2010. 1998 refers to a review of that book.
Done.

Line 95 and Eq. (4): Why is this equation explained and written in such a complicated way? For me it looks like to simply be density times volume for the hydrometeors considered: $m_x(D) = \rho_x V_x = \pi/6 \rho_x D^3$
We agree with the reviewer but we would like to highlight the density approximation used in the MY05 microphysics scheme.

Line 124 -124: You only consider homogeneous ice nucleation of pure supercooled water droplets. What about haze droplets and the resulting freezing point depression?
Indeed, homogeneous ice nucleation is possible for haze droplets but only for high value of $S_i$. For instance, using Barahona and Nenes (2009) parameterization, at a temperature of -40°C, $S_i$ have to be superior to 1.46 for the homogenous freezing to occur. Vertical profiles of temperatures and relative humidity over ice for the three simulated cases (see Fig. (3) and (4) in Keita et al. (2019)) show that $S_i$ is almost always under this threshold value both for simulated results and observations.

Eqs. (20) and (21) and Fig.1: Could you please make clear in the text, when introducing Eqs. (20) and (21), that Eq. (20) belongs to MYKE2 and Eq. (21) to MYKE4? It is mentioned in Fig. 1 but not in the text at the end of section 2.1.1.

The sentence "Both formulations are implemented in MY05 and tested hereafter." at Line 217 was changed for:

"Both formulations referred to MYKE2 (Eq. (20)) and MYKE4 (Eq. (21)) are implemented in MY05 and tested hereafter."

Cited:

Barahona, D. and Nenes, A.: Parameterizing the competition between homogeneous and heterogeneous freezing in cirrus cloud formation – monodisperse ice nuclei, Atmospheric Chemistry and Physics, 9, 369–381, doi:https://doi.org/10.5194/acp-9-369-2009, 2009.

Bigg, E. K.: The formation of atmospheric ice crystals by the freezing of droplets, Royal Meteorological Society, 79, 510-519, 1953.

Chen, J.-P., Hazra, A., and Levin, Z.: Parameterizing ice nucleation rates using contact angle and activation energy derived from laboratory data, Atmos. Chem. Phys., 8, 7431–7449, https://doi.org/10.5194/acp-8-7431-2008, 2008.

Eastwood, M. L., Cremel, S., Gehrke, C., Girard, E., and Bertram, A. K.: Ice nucleation on mineral dust particles: Onset conditions, nucleation rates and contact angles, Journal of Geophysical Research, 113, https://doi.org/10.1029/2008jd010639, 2008.

Eastwood, M. L., Cremel, S., Wheeler, M., Murray, B. J., Girard, E., and Bertram, A. K.: Effects of sulfuric acid and ammonium sulfate coatings on the ice nucleation properties of kaolinite particles, Geophysical Research Letters, 36, https://doi.org/10.1029/2008gl035997, 2009.

Ferrier, B. S.: A Double-Moment Multiple-Phase Four-Class Bulk Ice Scheme. Part I: Description, J. Atmos. Sci., 51(2), 249–280, doi:10.1175/1520-0469(1994)051<0249:ADMMPF>2.0.CO;2, 1994.

Fletcher, N. H.: Size effect in heterogeneous nucleation, Journal of Chemical Physics, 29, 572-576, 1958.

[revised manuscript text omitted]

---

## Author Comment (AC2) · 5 Sep 2020

The authors thank Reviewer#2 for this comprehensive review of the paper. We address below each comment individually (in blue color). Line numbers refer to the original manuscript.

My main concern is related to the lack of a proper justification for the proposed parameterizations. The authors base their development on CNT which accuracy for heterogeneous ice nucleation is still matter of debate, although it has been applied before. However the authors make some assumptions that need to be justified. Ice nucleation is assumed to occur mainly in the deposition mode or by immersion in solution. As mentioned by another reviewer only expressions for deposition ice nucleation are used. Moreover, why are these considered the main paths of ice nucleation in the stratiform clouds? Droplet freezing is probably more significant. If not, the authors should show some evidence or at least reports suggesting otherwise. Also, a control simulation where CNT is used but with no acidity dependency considered should be added to discriminate the effect of the later.

We thank Reviewer#2 for this comment, similar to points stressed by Reviewer#1. We copy here the detailed answer to that comment.

Our objective in developing the new parameterization was to represent the formation of ice crystals in the particular conditions of Arctic TIC clouds. In these conditions, it is mainly the deposition mode that occurs for the heterogeneous nucleation of ice, i.e. the air mass is in water-subsaturated regime. Kulkarni et al. (2014) showed that, except for quartz, acid-coated dusts are less effective INPs in the deposition mode but have similar effectiveness in the immersion-freezing mode, i.e. in water-supersaturated regime. Based on X-ray diffraction analyses, they argued that acid treatment caused structural deformations of the surface dusts, and the lack of structured order reduced the ice nucleation properties of coated particles in the deposition mode. Moreover, they suggested that, at water-supersaturated conditions, surface chemical reactions might not change the original ice nucleating properties permanently because coating material could be removed by dissolution. For kaolinite, Panda et al. (2010) concluded that sulfuric acid-treated particles could result in the formation of aluminum sulfate that can be easily dissolved in water. Considering these recent findings, and our objective to develop a simplified parameterization to limit computational time, we chose to use the CNT formula for deposition mode but with a specific factor, the neutralization fraction, indicating the degree of acidity of the coating of dust particles. Several passages of the text have been modified to clarify the conditions of the parameterization:

Line 55: "In Keita et al. (2019), the parameterization of Girard et al. (2013) for water-subsaturated conditions based upon CNT approach was implemented in the online Weather Research and Forecasting model coupled with chemistry (WRF-Chem) (Grell et al., 2005). This parameterization is suitable to represent the formation of ice clouds in Arctic. It assumes that INPs are mainly mineral dust particles, which is consistent with recent results from the NETCARE (Network on Climate and Aerosols: Addressing Key Uncertainties in Remote Canadian Environments) project (Abbatt et al., 2019)."

Line 66: "In this paper, we investigate for the first time the ice heterogeneous nucleation in a fully coupled aerosol and chemistry parameterization."

Line 78: "The new scheme for ice crystals formation by heterogeneous nucleation in the deposition mode is implemented…"

Line 153: "Moreover, the condensation-freezing mode, as discussed in Vali et al. (2015), is quite uncertain."

Line 157: "The new parameterization focuses on the heterogeneous ice nucleation for uncoated INPs and for sulphuric acid coated INPs in the deposition mode i.e. in water-subsaturated conditions. In this approach, INPs are assumed to be mineral dust particles following Girard et al. (2013). For contact freezing and immersion freezing from supercooled cloud droplets, the parameterizations remain unchanged. As condensation-freezing is uncertain (Vali et al., 2015), this process is not longer included in the model."

Line 207: "For instance, Kulkarni et al. (2014) showed that, except for quartz, acid-coated dusts are less effective INPs in the deposition mode but have similar effectiveness in the immersion-freezing mode, i.e. in water-supersaturated regime. Based on X-ray diffraction analyses, they argued that acid treatment caused structural deformations of the surface dusts, and the lack of structured order reduced the ice nucleation properties of coated particles in the deposition mode. Moreover, they suggested that, at water-supersaturated conditions, surface chemical reactions might not change the original ice nucleating properties permanently because coating material could be removed by dissolution. Panda et al. (2010) concluded that sulfuric acid-treated kaolinite particles could result in the formation of aluminium sulfate that can be easily dissolved in water. Considering these recent findings, and our objective to develop a simplified parameterization to limit computational time, we choose to use the CNT formula for deposition mode but with a specific factor, the neutralization fraction $f_n$, indicating the degree of acidity of the coating of dust particles."

Please, note that the immersion freezing of raindrops and cloud water droplets still follows the parameterization of (Bigg, 1953) but is not activated due to the absence of liquid drops in the simulated TIC clouds, except for some few exceptions in the lower part of clouds.

The authors have already performed several control simulations where CNT is used but with no acidity dependency, i.e. with a prescribed contact angle. Those results have been presented in Keita et al (2019) and compared to the same vertical cloud profiles obtained during ISDAC. The simulated vertical profiles of IWC, $R_i$ and $N_i$ found in Keita et al. (2019) for a contact angle of 12° or 26° turn out to be extreme cases of the new profiles described in the current paper. The new parameterization based on prognostic aerosols from WRF-Chem has the ability to distinguish polluted and non polluted air masses in the Arctic and to assess the ice crystal nucleation rate with a contact angle between 12° (clean air mass) and 26° (acidic air mass).

Lines 16-19. Please split this sentence.
Done

Line 26. Should be "specific"'.
For the sake of clarity, this paragraph has been thoroughly revisited:
 "The detailed process of ice nucleation in cold clouds is complex and remains a major challenge for parameterization in atmospheric models. This is especially the case for polar ice clouds, where the paucity of observations is a serious limitation (Curry et al. 1996; Kanji et al. 2017; McFarquhar et al. 2017). For instance, instead of assuming that cloud particles are distributed homogeneously, to investigate model response and climate sensitivity, some models have based their parameterization on in situ observations (Kay et al., 2016, Cirisan et al, 2020). However, the strong coupling between clouds and state variables, particularly temperature and moisture or relative humidity, requires a dynamic coupling of the cloud microphysics interactively with the atmospheric state variables.

Among these coupling processes, the efficiency of ice nuclei particles (INPs) to activate cloud formation is critical, given the rarity of INPs in the pristine atmosphere. Two approaches are used to treat the INPs efficiency; a singular and deterministic method, or a stochastic method (Pruppacher and Klett, 1997). While the singular approach assumes nucleation to occur at specific relative humidity and temperature (e.g. Wheeler and Bertram 2012; Murray et al. 2012), the stochastic method allows for time-dependent state variables following the classical nucleation theory (CNT) (Pruppacher and Klett, 1997; Cirisan et al, 2020). It is also our approach in this study, where we assume that freezing occurs at any location on the INP surface with equal probability. This is one attempt to represent best in situ observations, yet still not fully physically comprehensive, but one exploration step. The ultimate general method is still a matter of intense research (Vali, 2014; Wright and Petters, 2013)."

Line 29. Remove the comma.
Please refer to the answer to line 26 above.

Line 33. Number density is however a function of temperature.
Please refer to the answer to line 26 above.

Line 34. CNT is not a requirement of the stochastic hypothesis. Please rephrase.
Please refer to the answer to line 26 above.

Lines 36-39. Most atmospheric models use time-independent formulations. In fact, all of these references correspond to time-independent formulations.
There was a typo here. We had written "time-dependent" instead of "time-independent". This has been removed in the revised version.

Line 41. Please explain the significance of the contact angle. Also isn't this the approach used in this work? A single contact angle, dependent on the acidity?
In the CNT model, a crucial fitting parameter is the contact angle ($\theta$), quantifying the wettability of a solid particle surface by ice via the Young-Dupré equation. It is generally described as a single contact angle for an entire aerosol population, which does not work well for predicting the fractions of INPs on dust aerosol or on particles that have heterogeneous surfaces (Hoose and Möhler, 2012). In this paper, the contact angle is a function of the neutralization fraction, which in turn depends on the variable aerosol composition. It has been precised in the revised version of the manuscript.

Line 45. Say INP (ice nucleating particle) instead of IN.
Done

Line 55. Is dust internally mixed with sulfuric acid?
Yes. In the model description, the MOSAIC module is briefly introduced: MOSAIC uses a sectional approach to represent aerosol size distributions by dividing up the size distribution for each species into several size bins (8 used in this paper) and assumes that the aerosols are internally mixed in each bin.

Line 103. Is this assumption appropriate for small ice particles?
Yes.

Line 133. Why is immersion freezing of cloud droplets (which is likely the dominant path of ice formation) not treated in a more rigorous way?
We thank the reviewer for this comment.
Please refer to the answer to your main concern above.

Line 159, Eq. 13. Is this the total surface area? Shouldn't this equation be weighted by the aerosol size distribution? Also, when applying this to the immersion case, shouldn't it be only valid for the dust particles immersed within the haze aerosol droplets?
Yes, Ad is the total surface area of the aerosol particles. The number concentration of nucleated ice crystals could have been computed per size bin, but it has not been done in this paper. As a consequence, the total number of aerosol particles is used and their total surface area takes into account a weighting by the size distribution. The parameterization is only valid for the deposition mode.

Line 170. This seems wrong. Is it maybe 10^26?
It was indeed a typo. We change 10^-26 to 10^26.

Line 174. The surface tension between ice and vapor is a function of temperature.
Also, this would be invalid for immersion within haze particles.
This is right but the formulation of the parameterization only refers to the deposition mode.

Line 176. This is not the expression for an infinite plane surface. This is in fact the expression for small INP when the size is comparable to the size of the ice germ.
We agree, this is the expression for a curved substrate. It has been corrected.

Line 203, Eq. 19. Is this for the dust particles internally mixed with sulfate and nitrate, or the overall composition? The latter would not seem very rigorous. Please explain.
Yes, in the MOSAIC aerosol module, dust particles are assumed to be internally mixed with sulfate, nitrate and ammonium. The other components of the aerosol composition are not of interest in this study.

Line 215-220. What is the rationale behind the proposed functional forms in Eqs. 20 and 21? Why would the contact angle depend on the acidity?
We thank Reviewer#1 for this comment.
Keita and Girard (2016), after analysing the slope between the nucleation rate and the saturation over ice for TIC1 and TIC2 clouds (cf. Fig. 16 in Keita and Girard (2016)) observed for a given $S_i$ that: (1) the slope is the largest for the smallest accessible contact angles; (2) the decrease of the slope with the increasing contact angle is very non-linear. These results are consistent with laboratory experiments (Sullivan et al., 2010) showing a rapid increase of the contact angle with acidity on coated IN. These results motivated us to parameterize the contact angle $\theta$ as a function of the aerosol neutralization fraction under a concave form. Simple concave functions follow power law: $\theta = 26 - 14 \times f_n^p$ with p larger than 1. We have chosen a quadratic (p=2, MYKE2 simulation) form for simplicity. We have besides added a sensitivity simulation (MYKE4) under a biquadratic form (p=4) for simplicity to test the influence of the exponent p on the concave form of the contact angle with the neutralization fraction.

Line 233. There are no equations 20a and 20b.
We have replaced them by Eq. 20 an Eq. 21.

Line 265. If ice nucleation occurs at cloud top why would it be on haze aerosol instead of cloud droplets immersed with dust?
Please see the response to your main concern above.

Line 285. Is this the total aerosol number for all species?
Yes.

Line 327. This is a confusing sentence? What do the authors mean by the same f?
We rephrase it by "Results obtained with the MYKE2 and MYKE4 using the same value of the neutralization fraction are very similar."

Line 349-350. Can you show this in a plot?
Here we just mention the general behaviour of the nucleation of ice crystals in the CNT as a function of the contact angle. The critical free energy is proportional to the reduction factor $f(cos\theta)$ (Eq. 15), a monotonic decreasing function of the cosine of the contact angle (Eq. 16). Since the cosine is also a monotonic decreasing function between 0° and 90°, the energy barrier is a monotonic increasing function of the contact angle. As a consequence, a smaller contact angle in the simulation tends to decrease the critical Gibbs free energy to form ice embryos (Eq.15), hence leading to a higher nucleation rate of ice crystals (cf. Fig. 16 in Keita and Girard (2016)) and higher IWC. This explains the differences between MYKE2 and MYKE4.

Lines 384-385. Please show this.
Here is the figure showing the RH$_i$ as a function of altitude (in pressure levels) for the different simulations on the three cases.
As these results were already shown in Keita et al. (2019) (cf. Figure 4), we choose to not show them again in the present manuscript.

[Figure]

Line 402-403. What about using no f, i.e., Just a fixed contact angle?

This has already been done in Keita et al (2019). The current paper presents the big advantage to calculate a contact angle that adjusts to the acidity of the air mass. The spatial and temporal heterogeneities of air masses and ice clouds are thus better represented.

Cited:

Bigg, E. K.: The formation of atmospheric ice crystals by the freezing of droplets, Royal Meteorological Society, 79, 510-519, 1953.

Hoose, C. and Möhler, O.: Heterogeneous ice nucleation on atmospheric aerosols: a review of results from laboratory experiments, Atmospheric Chemistry and Physics, 12, 9817–9854, https://doi.org/10.5194/acp-12-9817-2012, 2012.

Keita, S., Girard, E., Raut, J.-C., Pelon, J., Blanchet, J.-P., Lemoine, O., and Onishi, T.: Simulating Arctic Ice Clouds during Spring Using an Advanced Ice Cloud Microphysics in the WRF Model, Atmosphere, 10, https://doi.org/10.3390/atmos10080433, 2019.

Keita, S. A. and Girard, E.: Importance of Chemical Composition of Ice Nuclei on the Formation of Arctic Ice Clouds, Pure and Applied Geophysics, 173, 3141–3163, https://doi.org/10.1007/s00024-016-1294-z, 2016.

Kulkarni, G., Sanders, C., Zhang, K., Liu, X. and Zhao, C.: Ice nucleation of bare and sulfuric acid-coated mineral dust particles and implication for cloud properties, Journal of Geophysical Research: Atmospheres, 119, 9993–10011, doi:10.1002/2014JD021567, 2014.

Panda, A. K., Mishra, B. G., Mishra, D. K., and Singh, R. K.: Effect of sulphuric acid treatment on the physico-chemical characteristics of kaolin clay, Colloids Surface, A, 363, 98–104, doi:10.1016/j.colsurfa.2010.04.022, 2010.

Sullivan, R. C., Petters, M. D., DeMott, P. J., Kreidenweis, S.M., Wex, H., Niedermeier, D., Hartmann, S., Clauss, T., Stratmann, F., Reitz, P., Schneider, J., and Sierau, B.: Irreversible loss of ice nucleation active sites in mineral dust particles caused by sulphuric acid condensation, Atmospheric Chemistry and Physics, 10, 11 471–11 487, https://doi.org/10.5194/acp-10-11471-2010, 2010.

Vali, G.: Interpretation of freezing nucleation experiments: singular and stochastic; sites and surfaces, Atmospheric Chemistry and Physics, 14, 5271–5294, https://doi.org/10.5194/acp-14-5271-2014, 2014.

---

## Author Comment (AC3) · 5 Sep 2020

The authors thank Reviewer#3 for this comprehensive review of the paper. We address below each comment individually (in blue color). Line numbers refer to the original manuscript.

The rational behind this new parameterization is not clearly presented. Section 2.1.1 (which should be section 2.2) must be rewritten. Why did the authors decide to change the nucleation rate and the contact angle, instead of another method? Why did they choose this relationship between the neutralized fraction and the contact angle ? How does this new parameterization fit in the Milbrandt and Yau scheme exactly (a diagram would help)? This section is confusing and incomplete.

Our objective in developing the new parameterization was to represent the formation of ice crystals in the particular conditions of Arctic TIC clouds. In these conditions, it is mainly the deposition mode that occurs for the heterogeneous nucleation of ice, i.e. the air mass is in water-subsaturated regime. Kulkarni et al. (2014) showed that, except for quartz, acid-coated dusts are less effective INPs in the deposition mode but have similar effectiveness in the immersion-freezing mode, i.e. in water-supersaturated regime. Based on X-ray diffraction analyses, they argued that acid treatment caused structural deformations of the surface dusts, and the lack of structured order reduced the ice nucleation properties of coated particles in the deposition mode. Moreover, they suggested that, at water-supersaturated conditions, surface chemical reactions might not change the original ice nucleating properties permanently because coating material could be removed by dissolution. For kaolinite, Panda et al. (2010) concluded that sulfuric acid-treated particles could result in the formation of aluminum sulfate that can be easily dissolved in water. Considering these recent findings, and our objective to develop a simplified parameterization to limit computational time, we chose to use the CNT formula for deposition mode but with a specific factor, the neutralization fraction, indicating the degree of acidity of the coating of dust particles."

Concerning the relationship between the contact angle and the neutralization fraction, Keita and Girard (2016), after analysing the slope between the nucleation rate and the saturation over ice for TIC1 and TIC2 clouds (cf. Fig. 16 in Keita and Girard (2016)) observed for a given $S_i$ that: (1) the slope is the largest for the smallest accessible contact angles; (2) the decrease of the slope with the increasing contact angle is very non-linear. These results are consistent with laboratory experiments (Sullivan et al., 2010) showing a rapid increase of the contact angle with acidity on coated IN. These results motivated us to parameterize the contact angle $\theta$ as a function of the aerosol neutralization fraction under a concave form. Simple concave functions follow power law: $\theta = 26 - 14 \times f_n^p$ with p larger than 1. We have chosen a quadratic (p=2, MYKE2 simulation) form for simplicity. We have besides added a sensitivity simulation (MYKE4) under a biquadratic form (p=4) for simplicity to test the influence of the exponent p on the concave form of the contact angle with the neutralization fraction.

We have been rewriting Sect. 2.2 taking into account the developed arguments above.

We choose not adding a diagram of Milbrandt and Yau scheme because we think that this is unnecessary with the new version of Sect. 2.2.

The paper clearly lacks proofreading. A lot of well-known and well-established equations contain mistakes.

The authors apologize for those typos and errors in the style. We have carefully corrected all of them.

Equation 1: velocity is missing in the first term of the right-hand side part of the equation, a dot is missing as well (convergence) ; the third term is d/dz and not d/dt
Done.

Equation 4: it is a PDF, therefore, Nx(D) = dn/dD, and writing dNx(D) does not make any sense. Ntx is the total number concentration, and is integrated over D, so it is Ntx and not Ntx(D). In the exponential, both lambda_x and D are to the power of nu_x, not only D, it is therefore (lambda_x D)^nu_x.
Done.

Equation 5: again, it is a PDF, and it is Nx(D) and not dNx(D).
Done.

Equation 6: Ntx and not Ntx(D)
Done.

Equation 8 is not consistent I believe; it is not in kg/kg, because of the 1/rho factor.
We removed the rho factor.

Equation 11: I don't understand where this equation comes from. Please demonstrate.
We just present here the original formulation to treat the homogeneous freezing of cloud droplets at temperature below -30 as in Milbrandt and Yau (2005a). All the details are presented in DeMott et al. (1994) and Milbrandt and Yau (2005b). According to Milbrandt and Yau (2005b), Eq. 11 is obtained by substituting the mean-droplet volume $\frac{\pi}{6} D_{mc}^3$ in Eq. 9.

We have rephrased line 127 by " with the volume V approximated by the mean volume, the fraction of cloud droplets freezing in one time step may be written as:" and we have added after the equation " where $D_{mc}$ is mean-droplet diameter".

Equation 16: usually Mw^2 also appears in the Gibbs free energy term;
The authors disagree with Reviewer#3. We have used Rv, the gas constant for water vapor (in J/kg/K). As a consequence, the molar mass of water is implicitly taken into account: Rv = Rg/Mw.

Equation 17: it is not q-qcos(theta) but q-cos(theta)
Done.

l.85: "All symbols for variables and parameters used are listed in Table 1." Where is Table 1 ? It appears to be missing. This probably explains why the numbering of all the other tables is wrong…
All Tables have been numbered again as Table 1 did not exist.

l.155: "For condensation-freezing, it can be included in the immersion freezing of coated IN when air is supersaturated with respect to liquid water." This sentence is quite confusing, and this whole paragraph is unclear. How does this new parameterization fit in the Milbrandt and Yau scheme exactly ? Please include a diagram, for example.
We thank Reviewer#3 for this comment. This paragraph has been thoroughly revisited.

"The parameterization for condensation-freezing can be derived from that of immersion freezing of coated INPs when air is supersaturated with respect to liquid water. Moreover, the condensation freezing mode, as discussed in Vali et al. (2015), is quite uncertain. The new parameterization focuses on the heterogeneous ice nucleation for uncoated INPs and for sulfuric acid coated INPs in the deposition mode, i.e. in water-subsaturated conditions. In this approach, INPs are assumed to be mineral dust particles following Girard et al. (2013). For contact freezing and immersion freezing from supercooled cloud droplets, the parameterizations remain unchanged. As condensation-freezing is uncertain Vali et al. (2015), this process is not longer included in the model."

Sections 4.2, 4.3, 4.4, 5: these sections are all made of one huge paragraph and are very hard to read.
For some reasons, line breaks splitting paragraphs were not indeed visible on the submitted version. Each Section is now clearly split in different coherent paragraphs.

l.606: the two references to Milbrandt and Yau are the same, and should be Part I and part II;
This was a mistake. It has been corrected.

Figure 6 is very hard to read.
We think that it is because of the legend in the box, which is confusing. We have changed it to clarify the figure and we hope that results are more readable. The new figure is reproduced below:

[Figure]

We have done the same modification of the legend for figures 3, 7 and 8.

Cited:

DeMott, P. J., Meyers, M. P. and Cotton, W. R.: Parameterization and Impact of Ice initiation Processes Relevant to Numerical Model Simulations of Cirrus Clouds, J. Atmos. Sci., 51(1), 77–90, doi:10.1175/1520-0469(1994)051<0077:PAIOII>2.0.CO;2, 1994.

Keita, S. A. and Girard, E.: Importance of Chemical Composition of Ice Nuclei on the Formation of Arctic Ice Clouds, Pure and Applied Geophysics, 173, 3141–3163, https://doi.org/10.1007/s00024-016-1294-z, 2016.

Kulkarni, G., Sanders, C., Zhang, K., Liu, X. and Zhao, C.: Ice nucleation of bare and sulfuric acid-coated mineral dust particles and implication for cloud properties, Journal of Geophysical Research: Atmospheres, 119, 9993–10011, doi:10.1002/2014JD021567, 2014.

Milbrandt, J. A. and Yau, M. K.: A Multimoment Bulk Microphysics Parameterization. Part I: Analysis of the Role of the Spectral Shape Parameter, J. Atmos. Sci., 62(9), 3051–3064, doi:10.1175/JAS3534.1, 2005a.

Milbrandt, J. A. and Yau, M. K.: A Multimoment Bulk Microphysics Parameterization. Part II: A Proposed Three-Moment Closure and Scheme Description, J. Atmos. Sci., 62(9), 3065–3081, doi:10.1175/JAS3535.1, 2005b.

Panda, A. K., Mishra, B. G., Mishra, D. K., and Singh, R. K.: Effect of sulphuric acid treatment on the physico-chemical characteristics of kaolin clay, Colloids Surface, A, 363, 98–104, doi:10.1016/j.colsurfa.2010.04.022, 2010.

Sullivan, R. C., Petters, M. D., DeMott, P. J., Kreidenweis, S.M., Wex, H., Niedermeier, D., Hartmann, S., Clauss, T., Stratmann, F., Reitz, P., Schneider, J., and Sierau, B.: Irreversible loss of ice nucleation active sites in mineral dust particles caused by sulphuric acid condensation, Atmospheric Chemistry and Physics, 10, 11 471–11 487, https://doi.org/10.5194/acp-10-11471-2010, 2010.

---

## Author Comment (AC4) · 17 Sep 2020

Dear executive editor,

Here the information regarding code and data availability.

Code and data availability.

WRF-Chem is an open-source community model. The source code of WRF-Chem model version 3.5.1 is available at http://www2.mmm.ucar.edu/wrf/users/download/get_source.html (last access: September 2020). The new scheme for ice crystals formation by heterogeneous nucleation described in this paper is implemented in WRF-Chem Version 3.5.1 and permanently archived at https://zenodo.org/badge/latestdoi/295455287 (last access: September 2020).

Indirect and Semi-Direct Aerosol Campaign (ISDAC) data are available from the ARM data archive (online at https://www.arm.gov/data/data-sources/cldmicroprop-51)

Meteorological initial and boundary conditions use NCEP (National Centers for Environmental Prediction) Global Forecast System (GFS) Final Analysis (FNL) data is available at https://rda.ucar.edu/datasets/ds083.2/ (last access: September 2020).

Chemical initial and boundary conditions are taken from the global chemical-transport model MOZART-4 (Model for OZone And Related chemical Tracers, version 4) (Emmons et al., 2010). https://www.acom.ucar.edu/wrf-chem/mozart.shtml (last access: September 2020).

The fire emissions inventory used is the Fire INventory from NCAR (FINN-v1) (Wiedinmyer et al., 2011) is available at http://bai.acom.ucar.edu/Data/fire/.

Aknowledgements : We acknowledge use of the WRF-Chem preprocessor tools mozbc and fire_emiss provided by the Atmospheric Chemistry Observations and Modeling Lab (ACOM) of NCAR.